# Increasing Probability Mass on Answer Choices
# Does Not Always Improve Accuracy

**Sarah Wiegreffe**♠ **Matthew Finlayson**◇* **Oyvind Tafjord**♠
**Peter Clark**♠ **Ashish Sabharwal**♠

♠Allen Institute for AI ◇University of Southern California
wiegreffesarah@gmail.com, mfinlays@usc.edu,
{oyvindt,peterc,ashishs}@allenai.org

## Abstract

When pretrained language models (LMs) are applied to discriminative tasks such as multiple-choice questions, they place probability mass on vocabulary tokens that aren't among the given answer choices. Spreading probability mass across multiple surface forms with identical meaning (such as "bath" and "bathtub") is thought to cause an underestimation of a model's true performance, referred to as the "surface form competition" (SFC) hypothesis. This has motivated the introduction of various probability normalization methods. However, many core questions remain unanswered. How do we measure SFC? Are there direct ways of reducing it, and does doing so improve task performance?

We propose a mathematical formalism for SFC which allows us to quantify and bound its impact for the first time. We identify a simple method for reducing it—namely, increasing probability mass on the given answer choices by a) including them in the prompt and b) using in-context learning with even just one example. We show this method eliminates the impact of SFC in the majority of instances. Our experiments on three diverse datasets and six LMs reveal several additional surprising findings. For example, both normalization and prompting methods for reducing SFC can be ineffective or even detrimental to task performance for some LMs. We conclude with practical insights for effectively prompting LMs for multiple-choice tasks.[1]

## 1 Introduction

Large pre-trained autoregressive language models (LMs) have shown success not only on generation, but also on classification and multiple-choice (MC) tasks with pre-specified answer choices. To succeed on such tasks, one must pay attention to

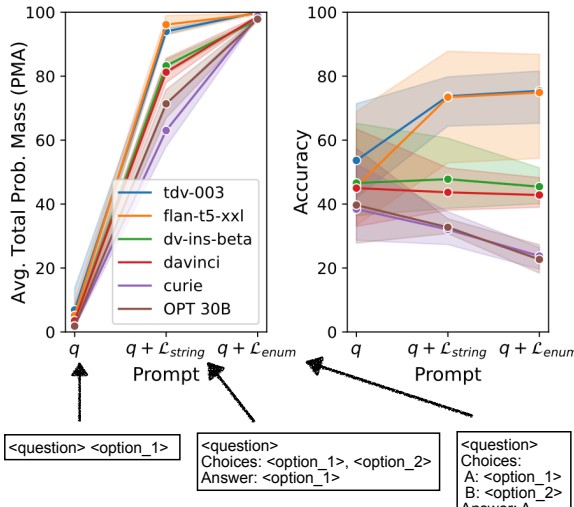

Figure 1: Higher probability mass on given answer choices (left; Eq. (5)) does not always translate to better accuracy (right; Eq. (1)), as shown here for three different prompt formats (x-axis; §6.3) each with one in-context example. Results are averaged across MMLU, OpenbookQA, and CommonsenseQA. Including answer choices in the prompt substantially increases probability mass on them. However, high probability mass is surprisingly **not always** associated with increased accuracy; in fact, it can lead to a **substantial drop** in performance (e.g., for OPT 30B and GPT-3 curie).

what's an acceptable answer choice[2] and what's not, i.e., understand the *task format*. This is accomplished relatively easily in the pretrain-and-finetune paradigm (Dai and Le, 2015; Howard and Ruder, 2018; Raffel et al., 2020; Lewis et al., 2020), via task-specific fine-tuning.[3]

However, in the zero- and few-shot prompting paradigm, in which the model is provided only a description or a handful of examples of the target task in the input, it's harder to ensure that the model

---

*Work done at AI2.
[1]Code available at https://github.com/allenai/revisiting_surface_form_competition.

[2]E.g., *True* and *False* in Boolean question-answering.

[3]For example, a T5 (Raffel et al., 2020) model fine-tuned on question-answering tasks (UnifiedQA; Khashabi et al., 2020) generates a given answer choice option on the OpenbookQA validation set 99.4% of the time.

generates only one of the answer choices associated with the given MC question. Most prior work tries to circumvent this issue by ignoring generated predictions and instead selecting the answer choice that has the highest probability under the model ("sequence scoring"; Trinh and Le, 2018; Radford et al., 2019; Brown et al., 2020, *i.a.*). This helps to some extent, by ignoring any attention the models pays to tokens unrelated to the answer choices. However, the problem persists as the model's probability mass can still be split among various strings or surface forms that are *semantically equivalent* to a given answer choice. Holtzman et al. (2021) propose that this phenomenon can result in underestimates of model performance, and refer to it as the **surface form competition** ("SFC") hypothesis. Motivated by this, they propose to use a *probability normalization* method, $PMI_{DC}$, to address the SFC issue, thereby (according to the SFC hypothesis) increasing model performance. In the same spirit, other probability normalization methods have been proposed (Zhao et al., 2021; Malkin et al., 2022), and their merit assessed via end task accuracy.

However, accuracy improvements may be attributable to multiple sources. Without a metric to directly measure SFC, it is difficult to assess whether the increased accuracy is, in fact, a consequence of reduced SFC, or something else.

To address this gap, we propose a mathematical formalism for studying SFC and use it to investigate the following four research questions.

**1. How can we measure SFC?** We propose to measure total probability mass on answer choices (abbreviated as PMA), and use it to upper bound the extent and impact of SFC (§4).

**2. How can we reduce SFC's effect?** Low PMA is a consequence of an inherently *under-constrained* output space that arises from the model failing to understand the task format. We use this observation to explain a simple way of **increasing PMA**: in-context learning with prompts containing answer choices (§5.1 and §5.2). We demonstrate the success of this approach across 6 LMs and 3 MC datasets (§7.1). We find, for instance, that when using this prompt with instruction-tuned LMs and just 2 in-context examples, on all 3 datasets, SFC simply couldn't have affected the prediction in more than 5% of instances.

**3. Does increasing PMA improve accuracy?** Surprisingly, not always! We provide an upper bound on the maximum effect an increase in PMA

can have on task accuracy (§4.1). We find empirically (Fig. 1 and §7.2) that the alignment between probability mass and accuracy isn't as clear cut as assumed in prior work (Holtzman et al., 2021)—it depends heavily on the model. These experiments also reveal that, contrary to common wisdom among researchers, encouraging models to produce answer choices by including them in the prompt can counter-intuitively be **detrimental** to task performance for LMs trained only on the next-token prediction objective.

**4. When do probability normalization methods improve accuracy?** While the direct effect of $PMI_{DC}$ on SFC is not easy to measure (§3.2), we extend prior work by studying when $PMI_{DC}$, which was motivated by SFC and is complimentary to our approach, improves accuracy on a wider set of prompts and models. We find, consistent with Holtzman et al. (2021), that $PMI_{DC}$ increases accuracy when models are *not* shown answer choices. However, this setting generally also corresponds to low probability assigned to answer choices. On the other hand, for the LMs that benefit from seeing answer choices, which results in high probability assigned to them, $PMI_{DC}$ scoring generally *reduces* accuracy. This indicates that as instruction-tuned LMs become more commonplace, PMI-based scoring methods, inspired by intuitions behind SFC (Holtzman et al., 2021), will likely provide less utility.

We conclude by leveraging these insights to provide practical recommendations on how to maximize LM accuracy on multiple-choice tasks when using zero- and few-shot prompting.

## 2 Related Work

While various methods have been proposed to improve the accuracy of sequence scoring using probability normalization methods (Brown et al., 2020; Zhao et al., 2021; Holtzman et al., 2021; Malkin et al., 2022), none investigate a direct metric for surface form competition and whether their methods alleviate it. To the best of our knowledge, we are the first to systematically study the role of in-context examples and prompt format on PMA, as well as how PMA relates to accuracy.

Holtzman et al. (2021) show $PMI_{DC}$ (Eq. (4)) improves over sequence scoring accuracy in most cases for GPT-2 and GPT-3 models of various sizes in 0-shot and 4-shot settings. Somewhat contradictorily, Brown et al. (2020) find that us-

ing a version of $\text{PMI}_{\text{DC}}$ where the denominator is $P_\theta(x|\text{"Answer : "}\text{or}\text{"A : "})$ improves task performance on the validation set for only 5 out of 17 datasets investigated. Zhao et al. (2021) propose to fit a linear weight matrix and bias vector for classification tasks with a shared label set, such that the labels all have equal probability prior to observing $x$. Malkin et al. (2022) add hyperparameters to Eq. (4) that are fit on a dataset's validation set, showing further gains at test-time. Min et al. (2022) propose to score inputs given answer choices, which is mathematically equivalent to $\text{PMI}_{\text{DC}}$ (§3.2). This results in lower variance and better *worst-case* accuracy on multiple-choice tasks in 0- and few-shot settings for GPT-2.

Liang et al. (2022) investigate the effect of showing answer choices in the prompt and applying PMI based scoring (though not the combination of the two). They find that the success of one method over the other tends to vary by dataset and model. Our results elucidate further that the overall capability of an LM may be a key factor in whether PMI-based scoring improves accuracy or not.

# 3 Preliminaries

Given a task input $x$, a set of answer choices $\mathcal{L}$, and the correct answer $y^* \in \mathcal{L}$, the goal of a multiple-choice classification task is to correctly select $y^*$. $x$ is often specified as a question $q$ and, optionally, answer choices $\mathcal{L}$ concatenated to $q$ as one string.[4]

Let $M$ be a generative model architecture with learned parameters $\theta$ and space of natural language outputs $\mathcal{V}$. In LMs, $|\mathcal{V}| \gg |\mathcal{L}|$, so generating a prediction $\hat{y}$ from $\mathcal{V}$ without constraints does not ensure that it is one of the given answer choices (i.e., that $\hat{y} \in \mathcal{L}$). Instead, we can use a *sequence scoring* approach to score each answer choice:

$$\hat{y}^{\text{Seq-Sc}} = \underset{\ell \in \mathcal{L}}{\text{argmax}} \, P_\theta(\ell|x) \quad (1)$$

where $P_\theta(y)$ is the probability $M_\theta$ assigns to output $y$.[5] This is a common approach for performing classification with generative LMs, as it ensures $\hat{y}^{\text{Seq-Sc}} \in \mathcal{L}$. This will be our prediction setup.

---

[4]For instance, if $x$ is a true/false question, $\mathcal{L}$ may be $\{True, False\}$. For a multiple choice question, $\mathcal{L}$ may be the set of (string) answers, their labels such as A/B/C/D, or both, depending on the format used to pose the task to an LM.

[5]For multi-token outputs $y = y_1 y_2 \ldots y_k$, we compute $P_\theta(y|x)$ as $\prod_{i=1}^{k} P_\theta(y_i|x, y_1 \ldots y_{i-1})$.

## 3.1 Surface Form Competition (SFC)

An LM's vocabulary contains many different strings, or *surface forms*, representing the same (or similar) semantic concept, but typically only one representative string per concept is among the given answer choices $\mathcal{L}$. Formally, for each answer choice $\ell \in \mathcal{L}$, there exists a set of synonyms $\mathcal{G}_\ell$ (containing $\ell$) that may be "stealing" probability mass away from $\ell$, while $\ell$ is the only surface form in $\mathcal{G}_\ell$ that is considered when computing accuracy via Eq. (1). One can quantify the amount of the resulting *surface form competition* as:

$$\text{SFC}_\theta(\mathcal{L}, x) = \sum_{\ell \in \mathcal{L}} \Big( P_\theta(\mathcal{G}_\ell|x) - P_\theta(\ell|x) \Big) \quad (2)$$

We refer to $\mathcal{G}_\ell$ as a *semantic equivalence class*, following Kuhn et al. (2023). For example, if $\mathcal{L} = \{A, B, C\}$, then there exist semantic equivalence classes $\mathcal{G}_A, \mathcal{G}_B$, and $\mathcal{G}_C$ containing all synonyms of $A, B$, and $C$, respectively. If $A = whirlpool\,bath$, $\mathcal{G}_A$ might be $\{whirlpool\,bath, bath, bathtub, \ldots\}$, which are all generations the LM may use to express the semantically similar concept.

The **SFC hypothesis** put forward by Holtzman et al. (2021) is that unresolved SFC results in an *underestimate* of model performance. They provide an example of how SFC may lead to incorrect predictions, which we extend and adapt in Fig. 2 (left): when LMs distribute probability mass across surface forms such that $P_\theta(y^*|x) < P_\theta(\ell|x)$ for some incorrect $\ell \in \mathcal{L}$, the model's prediction will be considered incorrect even if the *total* probability placed by the model on the correct *concept* $\mathcal{G}_{y^*}$ is higher than what it places on incorrect concepts.

To circumvent this issue, one may consider the notion of an **"SFC-free" prediction**: compute the most likely option among semantic equivalence classes, rather than among specific surface forms:

$$\hat{y}^{\text{SFC-free}} = \underset{\ell \in \mathcal{L}}{\text{argmax}} \, P_\theta(\mathcal{G}_\ell|x) \quad (3)$$

where $P_\theta(\mathcal{G}_\ell|x) = \sum_{z \in \mathcal{G}_\ell} P_\theta(z|x)$. A limitation of this formulation, however, is that it is only possible to compute $\hat{y}^{\text{SFC-free}}$ if the full membership of each $\mathcal{G}_\ell$ is known, which is rarely the case. LM vocabularies typically contain many tens of thousands of tokens, many of which may be partial synonyms.[6] This motivates the need for practical workarounds.

---

[6]Kuhn et al. (2023) use sampling and unsupervised clustering to approximate $P_\theta(\mathcal{G}_\ell|x)$. This approximation, however, is noisy by nature. They use it only to estimate semantic uncertainty, not to improve task accuracy.

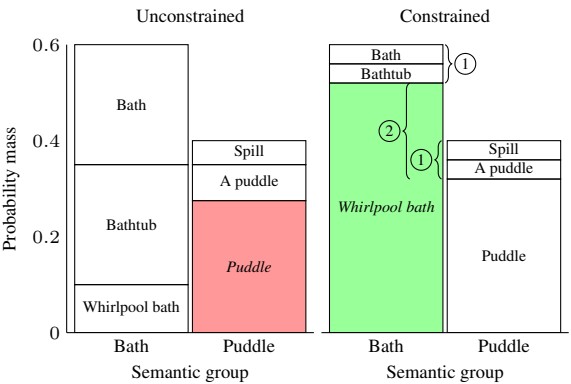

Figure 2: Left: A visualization of the SFC hypothesis: strings that are not answer choices can "steal" probability from the correct answer choice ("whirlpool bath"), leading to an incorrect prediction ("puddle"). Right: LMs can be constrained to place more probability mass on the answer choices (§5). So long as the probability mass on other strings ① is less than the difference in probability mass between the top two answer choices ②, SFC cannot affect a model's prediction (§4.1).

### 3.2 $\text{PMI}_{\text{DC}}$ as a Workaround

Holtzman et al. (2021) propose the following alternative selection method, $\text{PMI}_{\text{DC}}$:[7]

$$\hat{y}^{\text{PMI-DC}} = \operatorname*{argmax}_{\ell \in \mathcal{L}} \frac{P_\theta(\ell|x)}{P_\theta(\ell)} \qquad (4)$$

Intuitively, $\text{PMI}_{\text{DC}}$ measures the causal effect[8] of the input $x$ on the probability assigned to each answer choice $\ell$, and selects $\hat{y}$ as the answer choice on which $x$ has the largest effect. The method is an alternative scoring function; it doesn't change the underlying probabilities of the LM, $P_\theta$.

It is unclear when $\hat{y}^{\text{PMI-DC}} = \hat{y}^{\text{SFC-free}}$, i.e., when Eqs. (3) and (4) lead to the same, SFC-free prediction. Holtzman et al. note that $\text{PMI}_{\text{DC}}$ is mathematically equivalent to $\operatorname*{argmax}_{\ell \in \mathcal{L}} P_\theta(x|\ell)$. This, in turn, should intuitively not be far from $\operatorname*{argmax}_{\ell \in \mathcal{L}} P_\theta(x|\mathcal{G}_\ell)$ when $\ell$ is not directly mentioned in the question (which is the setting used in $\text{PMI}_{\text{DC}}$). In this case, the competition among surface forms within $\mathcal{G}_\ell$ would be alleviated. However, there is still no *a priori* reason for either $\operatorname*{argmax}_{\ell \in \mathcal{L}} P_\theta(x|\ell)$ or $\operatorname*{argmax}_{\ell \in \mathcal{L}} P_\theta(x|\mathcal{G}_\ell)$ to be the same as $\hat{y}^{\text{SFC-free}}$. Moreover, this view reveals a

---

[7]W.l.o.g., we ignore their use of a "domain context" string in the denominator. See Appendix A.4.

[8]In the sense that it measures the multiplicative factor by which the probability of $\ell$ increases upon observing $x$.

different competition, namely, among various *questions* $x$ whose answer (according to the model) is $\ell$. Specifically, a choice that the model thinks is more "popular" (i.e., the answer to *many* questions) will receive an artificially lower $\text{PMI}_{\text{DC}}$ score. Thus, now different questions (rather than different surface forms) compete for each answer choice.

## 4  How can we measure SFC?

Prior work has solely used the task accuracy metric to evaluate approaches geared towards resolving SFC. However, it is unclear whether task accuracy is an effective measure of the amount of SFC present. In fact, as we will show later, task accuracy is often *not* correlated with the amount of SFC.

While it is difficult to measure SFC (Eq. (2)) directly, we propose bounding it by considering the model $M_\theta$'s **probability mass on answer choices** or PMA, defined as follows:

$$\text{PMA}_\theta(\mathcal{L}, x) = \sum_{\ell \in \mathcal{L}} P_\theta(\ell|x) \qquad (5)$$

We assume no surface form in $\mathcal{L}$ is a prefix of another, in which case $\text{PMA}_\theta(\mathcal{L}, x) \leq 1$ (see Appendix A.5 for a proof and empirical verification). Intuitively, if a model is properly trained or instructed, it would place all probability mass on $\mathcal{L}$, resulting in $\text{PMA}_\theta(\mathcal{L}, x) = 1$. However, if SFC exists, we would observe $\text{PMA}_\theta(\mathcal{L}, x) < 1$.

Combining Eqs. (2) and (5) and observing that $\sum_{\ell \in \mathcal{L}} P_\theta(\mathcal{G}_\ell|x) \leq 1$, we obtain a bound on SFC:

$$0 \leq \text{SFC}_\theta(\mathcal{L}, x) \leq 1 - \text{PMA}_\theta(\mathcal{L}, x) \qquad (6)$$

### 4.1  When Can SFC Impact Accuracy?

The formulation of SFC as a measurable quantity enables quantifying the maximum amount by which it may impact a prediction. Specifically, the probability mass that does *not* fall on $\mathcal{L}$ cannot affect the model's final prediction if it is less than the difference in probability between the highest-probability answer choice, $\hat{y}$, and the second-highest-probability answer choice, $y_2 \in \mathcal{L}$. The right-hand side of Fig. 2 illustrates this principle. For example, if the probability of $\hat{y}$, "whirlpool bath", is $0.55$ and the probability of $y_2$, "puddle", is $0.35$, then $\text{PMA} = 0.9$ and the remaining probability mass is $0.1$. Even if all of this remaining probability mass were on synonyms of "puddle", the probability of "puddle" would only increase to $0.45$ should SFC be fully resolved, which would not flip the prediction since it is still less than $0.55$.

Combining this observation with Eq. (6), SFC simply *cannot* affect the output of $M_\theta$ on $x$ when:

$$1 - \text{PMA}_\theta(\mathcal{L}, x) \; < \; P_\theta(\hat{y}|x) - P_\theta(y_2|x) \quad (7)$$

Thus, one can completely remove the impact of SFC on a model's accuracy (i.e., achieve $\hat{y}^{\text{Seq-Sc}} = \hat{y}^{\text{SFC-free}}$) by raising PMA high enough relative to the gap between the probabilities of $\hat{y}$ and $y_2$; SFC doesn't have to be fully resolved (i.e., one need not push all the way to $\text{PMA} = 1$).

## 5 How can SFC be reduced?

The quantities used in $\text{PMI}_{\text{DC}}$ do not represent a valid probability distribution as they may exceed 1,[9] making it difficult to compute our proposed metric $\text{PMA}_\theta(\mathcal{L}, x)$. Is there a more straightforward way to equate $\hat{y}^{\text{Seq-Sc}}$ and $\hat{y}^{\text{SFC-free}}$?

### 5.1 Using In-Context Examples

One path forward is to somehow directly constrain the model $M_\theta$ such that $P_\theta(\mathcal{G}_\ell|x) = P_\theta(\ell|x)$ for all $\ell \in \mathcal{L}$, i.e., ensure that the answer choice $\ell$ is the only synonym in $\mathcal{G}_\ell$ to which $M_\theta$ assigns a non-zero probability mass. This, we posit, will occur naturally when LMs are properly constrained or instructed (see Fig. 1, left plot, right point).

One means to achieve this is to condition the predictions of $M_\theta$ on not only $x$ but also on some in-context examples $e_0, \dots, e_k$: $\hat{y}^{\text{ICE}} = \operatorname{argmax}_{\ell \in \mathcal{L}} P_\theta(\ell|x; e_0, \dots, e_k)$. Given that in-context examples are already widely used in practice, this technique is simple and straightforward to implement. Additionally, it allows one to easily compute $\text{PMA}_\theta(\mathcal{L}, x)$ for measuring the extent of SFC. In §6, we demonstrate empirically that with effective conditioning (prompt format and number of in-context examples), using in-context examples can significantly reduce SFC, and sometimes even *completely* resolve it by satisfying Eq. (7).

### 5.2 Prompting With Answer Choices

A key design decision when choosing which format to use to specify $x$ (and optionally in-context examples $e_0, \dots, e_k$) is whether to provide the model only the question $q$ or also the answer choices $\mathcal{L}$. Our PMA metric can be used to provide insight into this, by helping disentangle the contribution that each of $q$ and $\mathcal{L}$ makes to the task accuracy as well as to reducing surface form competition.

Intuitively, conditioning the prediction on $\mathcal{L}$ makes the model aware of what's an answer choice and what's not. It can thus push the model towards the specific surface forms contained in $\mathcal{L}$, without necessarily affecting model accuracy. This, by definition, directly increases the probability mass on answer choices. One can empirically quantify the effect of exposure to $\mathcal{L}$ by considering the gain one observes in PMA and in accuracy when going from $P_\theta(\ell)$ to $P_\theta(\ell|\mathcal{L})$.

On the other hand, one would expect that conditioning the prediction on $q$ pushes the model towards the correct semantic concept, i.e., the semantic equivalence class $\mathcal{G}^*$ of the correct answer. However, not knowing which specific surface form $\ell^*$ appears in both $\mathcal{G}^*$ and $\mathcal{L}$, the model has no reason to prefer $\ell^*$ over other equivalent surface forms $\ell \in \mathcal{G}^* \setminus \{\ell^*\}$. Thus, conditioning on $q$ alone can increase accuracy by increasing the probability mass on $\mathcal{G}^*$, but it does not resolve SFC within $\mathcal{G}^*$. We can, again, measure this by considering the gain in PMA and accuracy when going from either $P_\theta(\ell)$ to $P_\theta(\ell|q)$ or from $P_\theta(\ell|\mathcal{L})$ to $P_\theta(\ell|q, \mathcal{L})$.

## 6 Experiments

### 6.1 Models

We experiment with 6 models, described below.

**Vanilla LMs** These are models that are (to the best of publicly-available knowledge) only trained on the next-token prediction task. We experiment on two GPT-3 base models (Brown et al., 2020)— curie (~6.7B parameters) and davinci (~175B parameters)— and one model whose weights are publicly available, OPT 30B (Zhang et al., 2022).[10]

**LMs with Further Fine-Tuning** We study two instruction-tuned (Mishra et al., 2022, *i.a.*) models: FLAN-T5 XXL (~11B parameters; Chung et al., 2022), and the "original" InstructGPT model, GPT-3 davinci-instruct-beta (~175B parameters; Ouyang et al., 2022). We additionally test one "state of the art" model, GPT-3 text-davinci-003 (unknown # parameters). FLAN-T5 is based on the T5 architecture (Raffel et al., 2020) and its weights are publicly

---

[9]The quantity $\frac{P_\theta(\ell|x)}{P_\theta(\ell)}$ can, in principle, be viewed as the *unnormalized* probability of $\ell$. However, turning it into a proper probability distribution requires computing the normalization factor $\sum_z \frac{P_\theta(z|x)}{P_\theta(z)}$, which is prohibitively expensive and also unreliable, since LMs are generally not well-calibrated on the long tail of low-probability tokens.

[10]Sizes and corresponding citations for GPT-3 models are approximate; see OpenAI (2022).

available. It has demonstrated comparable performance to GPT-3 davinci despite being ~16x smaller. We include davinci-instruct-beta to study the effect of supervised instruction tuning on a model of identical scale to davinci-base that is also associated with a publicly-available research paper.[11] text-davinci-003 is (along with text-davinci-002) a state-of-the-art model according to the HELM benchmark (Liang et al., 2022). See Appendix A.1 for further details.

## 6.2 Tasks

We test on three challenging multiple-choice tasks that are open-vocabulary (i.e., each instance has a unique set of answer choices). Examples of the tasks are given in Appendix A.3; see also A.2.

**OpenbookQA** (Mihaylov et al., 2018) is a 4-way multiple-choice elementary-level science question-answering task. Random accuracy is 25%. The test set has 500 instances. **CommonsenseQA v1.11** (Talmor et al., 2019) is a 5-way multiple-choice commonsense reasoning task. Random accuracy is 20%. The test set is not publicly available; we use the first 500 instances of the validation set. Both OpenbookQA and CommonsenseQA were explicitly included in the training data of FLAN-T5. **MMLU** (Hendrycks et al., 2021), or the "Massive Multitask Language Understanding" benchmark, spans 57 different topical areas. The questions are 4-way multiple-choice spanning subjects in social sciences, STEM, and humanities that were manually scraped from practice materials available online for exams such as the GRE and the U.S. Medical Licensing Exam. Many state-of-the-art models perform poorly (random accuracy is 25%). We evaluate on the first 20 test questions from each category (1140 instances total).

## 6.3 Prompts

**In-Context Examples** We experiment with $k = 0, 1, 2, 4$ and $8$ in-context demonstrations, which are the same for each instance, and selected as the first $k$ examples from a fixed set of 8. For curie, davinci, and davinci-instruct-beta models, we report the mean and standard error over 3 random seeds used to select the set of 8 demonstrations, since the choice of in-context demonstrations can significantly affect performance (Lu et al., 2022, i.a.). We select in-context examples from

---

[11]It has been hypothesized that davinci-instruct-beta has been tuned directly from davinci checkpoint (Fu, 2022).

each dataset's associated training set (combined dev + validation sets for MMLU).

**Prompt Format** We experiment with three prompt formats, corresponding to the format of $x$ in §5.2. The first, "$q$", only contains the question and is thus most similar to next-word prediction. This is identical to the prompt used by Brown et al. (2020). For example,

```
kinetics change stored energy into
```

The second format, "$q + \mathcal{L}_{string}$", includes answer choices as a string list and is similar to formats included in PromptSource (Bach et al., 2022) used to train FLAN-T5 and other models:

```
question: kinetics change stored energy into
answer choices: snacks, naps, kites, or warmth
The correct answer is:
```

For both of the above prompts, models score or output the full string answer, e.g., warmth. The third format, "$q + \mathcal{L}_{enum}$", includes enumerated newline answer choices, similar to that used for zero-shot evaluations in FLAN (Wei et al., 2022) and FLAN-T5:

```
Question: kinetics change stored energy into
Choices:
 A: snacks\n B: naps\n C: kites\n D: warmth
Answer:
```

Here, models score only the (single-token) symbols, e.g., D. The full prompts are given in Appendix A.3, and their $\text{PMI}_{\text{DC}}$ denominators in A.4.

## 7 Results and Discussion

### 7.1 On Reducing Surface Form Competition

Fig. 1 (left), demonstrates the effect of choice of prompt format on PMA (and hence SFC, Eq. (6)) in the one-shot setting. Across datasets, showing answer choices in the "string" format leads to a substantial increase in PMA, which reaches near-100% for all models using the "$q + \mathcal{L}_{enum}$" format. Zooming in on the role of in-context examples in Fig. 3 (dashed lines), we observe PMA increases significantly for all models after seeing only one in-context example *that includes the answer choices* (bottom plot), and stronger models such as text-davinci-003 and FLAN-T5 exhibit this behavior zero-shot. Trends also hold for CommonsenseQA and OpenbookQA (Figs. 9 and 10, Appendix). The number of instances for which the bound in Eq. (7) is satisfied and SFC is fully alleviated are in Table 8 (Appendix A.6).

## 7.2 Relationship between Surface Form Competition and Accuracy

Fig. 1 (right), demonstrates the effect of the choice of prompt format on accuracy in the one-shot setting. While gains in PMA are consistent across models, this is not the case for accuracy. Certain models (curie, OPT 30B) actually achieve their best task performance when their PMA is the *lowest*, perhaps due to the $q$ prompt being the closest to the next-token prediction objective. For others (davinci, davinci-instruct-beta), accuracy is stable across prompts, even while PMA substantially increases. Seeing the answer choices in the prompt is crucial to achieving good accuracy with text-davinci-003 and FLAN-T5, likely due to their instruction tuning. Thus, showing answer choices does **not guarantee** improved accuracy, especially for vanilla LMs.

We can also observe this lack of positive correlation from the angle of in-context examples (Figs. 3, 9 and 10). While PMA increases with more in-context examples, accuracy is relatively stable across all models and prompt formats.

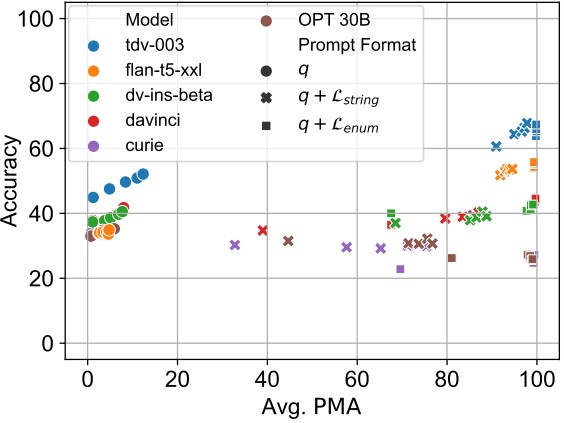

Figure 4: The relationship between task accuracy and average PMA (Eq. (5)) for the MMLU test subset (for 0, 1, 2, 4, and 8 in-context examples). See Fig. 7 in Appendix A.6 for CommonsenseQA and OpenbookQA, and Table 1 for Spearman's correlations.

Fig. 4 shows a shared scatterplot where each datapoint is a model result. The graph further illustrates the lack of correlation between increases in PMA (x-axis) and increases in accuracy (y-axis), especially in the bottom portion where PMA increases without any shift in y-axis position. In Table 1, we observe further evidence that PMA and accuracy are very negatively correlated in the case of curie and OPT 30B, and very positively correlated in the case of FLAN-T5 and text-davinci-003. davinci and davinci-instruct-beta exhibit highly variable correlation, indicating that the choice of LM modulates the PMA-accuracy relationship.

### 7.2.1 Role of Different Parts of the Input

In Fig. 5, we follow the methodology proposed in §5.2 and break down the zero-shot contributions to probability mass and accuracy of question $q$ vs. answer choices $\mathcal{L}$ when included in the prompt.

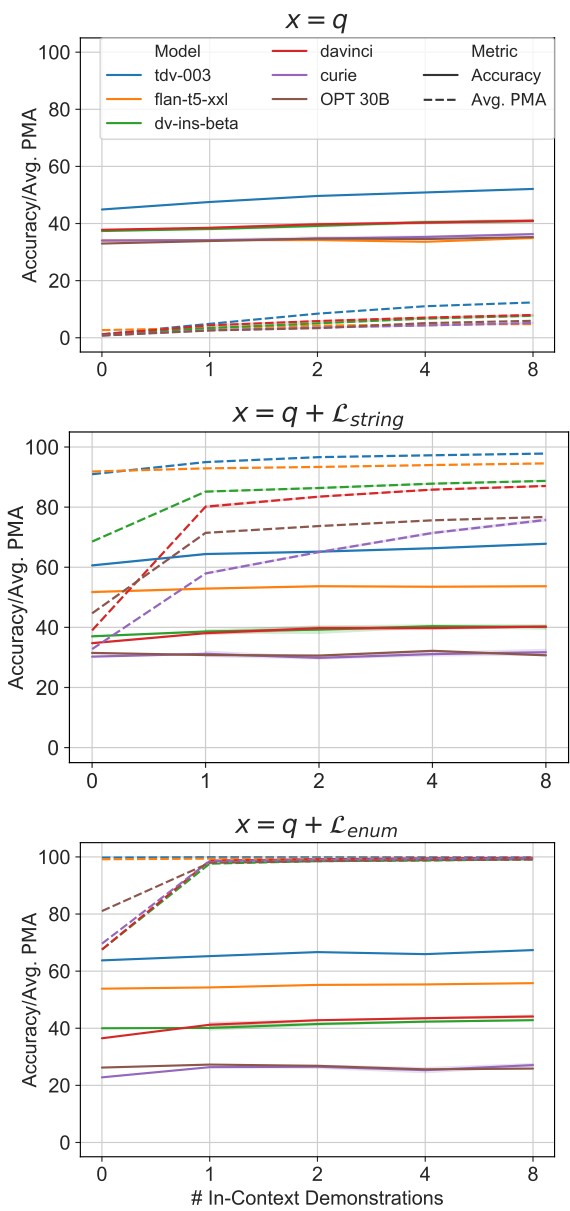

Figure 3: MMLU test set accuracy (Eq. (1); solid lines) and average PMA (Eq. (5); dashed lines) as a function of **number** (x-axis) and **format** (by graph) of in-context examples, for six pretrained LMs.

| Dataset | **Model** | | | | | |
|---|---|---|---|---|---|---|
| | curie | OPT 30B | davinci | davinci-instruct-beta | FLAN-T5 | text-davinci-003 |
| MMLU | $-0.84$ | $-0.84$ | 0.45 | 0.47 | **1.00** | **0.98** |
| CommonsenseQA | $-0.88$ | $-0.91$ | $-0.62$ | $-0.63$ | **1.00** | **0.86** |
| OpenbookQA | $-0.50$ | $-0.42$ | **0.78** | **0.84** | **1.00** | **1.00** |

Table 1: Per-model Spearman's correlations between avg. PMA and accuracy (as plotted in Figs. 4 and 7). **Bold**: results are statistically significant at $p < 0.05$ for a two-sided hypothesis test.

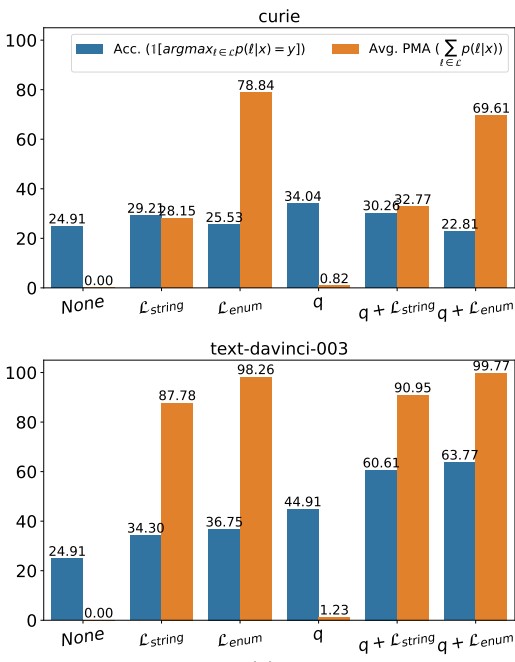

Figure 5: Zero-shot results on the MMLU test set: accuracy (Eq. (1); blue) and PMA (Eq. (5); orange) averaged over dataset instances. Observing answer choices in the prompt contributes far more to PMA than observing the question, confirming our hypothesis in §5.2. Even without observing the question, all models place a substantial amount of probability mass on answer choices after observing them in the prompt (see Fig. 11 for other models, and Appendix A.3 for prompt details).

We find that conditioning $P_\theta(\ell)$ on $\mathcal{L}$ (i.e., considering $P_\theta(\ell|\mathcal{L})$) substantially increases PMA (67.23% vs. 0% PMA on average for MMLU; the accuracy of both is similar, at 24.17% and 30.1%, resp. (5.93% absolute gain)). On the other hand, conditioning either of these probabilities further on $q$ (i.e., considering $P_\theta(\ell|q)$ or $P_\theta(\ell|q,\mathcal{L})$) provides a very small gain on PMA (2.97% absolute) as opposed to 11.88% accuracy gain on average for MMLU. This indicates that conditioning on $q$ is not an effective way to increase PMA (or decrease SFC). Overall, observing $q$ plays a larger role on accuracy while observing $\mathcal{L}$ plays a larger role in increasing PMA. In other words, observing $q$

appears to raise the *relative* probability of $y^*$ by redistributing mass among the members of $\mathcal{L}$, while observing $\mathcal{L}$ helps to raise the *absolute* probability given to $\mathcal{L}$ (i.e., PMA). Results hold for other models and datasets (Figs. 11 to 13).

### 7.3 When does $\mathrm{PMI_{DC}}$ improve accuracy?

Our experiments provide further insight into when normalization methods like $\mathrm{PMI_{DC}}$ may succeed. Fig. 6 (also Fig. 8) illustrates how much $\mathrm{PMI_{DC}}$ affects accuracy for each dataset.

Whether $\mathrm{PMI_{DC}}$ improves accuracy for a model seems tied to the largest PMA achieved by some prompt for that model as well as the model's overall performance: lower PMA and lower accuracy imply higher gains from $\mathrm{PMI_{DC}}$. Indeed, $\mathrm{PMI_{DC}}$ *always* improves accuracy when answer choices are not observed in the prompt (Figs. 6b and 8), and the extent of gain is fairly consistent for each dataset across number of in-context examples and models. However, as established earlier, prompting without answer choices often results in the worst accuracy for strong models. Fig. 6a plots the difference between the best accuracies using each method; gains are relatively muted, except for OpenbookQA. Additionally, $\mathrm{PMI_{DC}}$ generally (though not always) leads to significant accuracy drops for the strongest models (text-davinci-003 and FLAN-T5).

Tabular results for all experiments are in Tables 9 to 11 (Appendix). For curie, davinci, and davinci-instruct-beta, we include standard error over 3 random seeds for example selection. The effects of random seed are generally negligible.

## 8 Conclusion

We take a novel approach to studying the effects of prompt format, in-context examples, and model type on probability assigned to answer choices and its relationship with end task performance, by proposing a new formalization of surface form competition and a quantifiable metric (PMA). This is

an important step towards understanding and improving the use of LMs for discriminative tasks. Our findings shed light into the role of probability distributions in model performance. They also challenge intuitive assumptions such as showing answer choices for MC tasks is always beneficial, which is a common practice (Hendrycks et al., 2021; Rae et al., 2021; Hoffmann et al., 2022, *i.a.*).

**Practical Insights:** We find that the best way to use vanilla LMs in multiple-choice settings is to provide a string prompt *without* answer choices and apply probability normalization. For instruction-tuned models, on the other hand, answer choices *should* be shown and in an enumerated prompt format, and probability normalization should *not* be used. More generally, our results reveal that

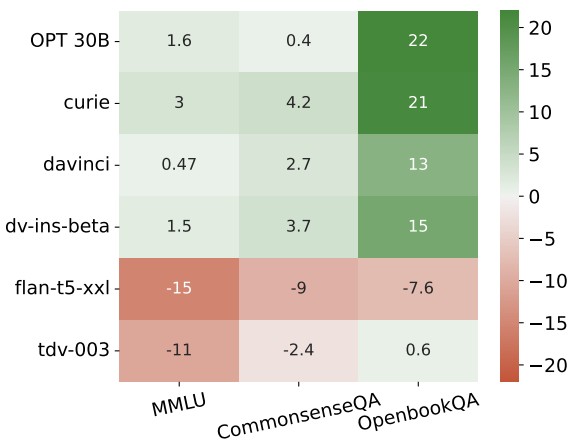

(a) Best across prompts

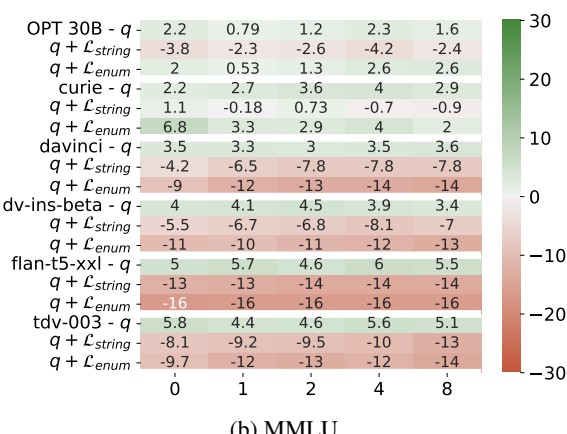

(b) MMLU

Figure 6: Accuracy changes achieved by using $\mathrm{PMI_{DC}}$ (Eq. (4)) over standard sequence scoring (Eq. (1)). Top: Differences in the best accuracy achieved by $\mathrm{PMI_{DC}}$ (Eq. (4)) and the best achieved by sequence scoring (Eq. (1)) across prompt settings for each model (y-axis) and dataset (x-axis). Bottom: full detail results for MMLU; other datasets are in Fig. 8.

efforts to increase probability assigned to answer choices via prompting methods can have surprisingly negative effects, and that scoring methods can drastically affect the conclusions we reach about an underlying LM's fundamental capabilities. We advocate future work to look into length normalization as another understudied scoring mechanism.

## Limitations

As with all papers using GPT-3 models, there is some stochasticity on the backend of the OpenAI API that researchers cannot control (studied in more depth by Ruis et al. (2022)). This means that results may vary from run to run, hampering reproducibility. In our setting, we find the effects to be very small in practice.

Additionally, in this work we only investigate open-vocabulary multiple-choice QA tasks. Future work might consider a broader suite of tasks or tasks where the answer choices are shared across instances, as in-context examples may have a larger effect on PMA or accuracy in that setting. Furthermore, we do not consider any directly comparable models for reaching conclusions about instruction tuning (base model → instruction-tuned) due to a lack of publicly available ones at the time this research was conducted; such an experiment would allow more concrete claims about the effect of instruction tuning and relationship with $\mathrm{PMI_{DC}}$ to be made.

Finally, there are other probability normalization variants that differ from standard $\mathrm{PMI_{DC}}$ in subtle ways (cited in §2). We only compare against the most straightforward (and common) implementation here.

## Ethics and Broader Impacts

This paper investigates the interplay between probability mass on vocabulary items and accuracy in zero-shot- and few-shot-prompted autoregressive language models. Our efforts show that investigations into output scoring functions can change the conclusions drawn about the capabilities of models, which we believe is an important part of better understanding how to reliably and adequately use these systems. The existing NLP benchmarks used both have limitations in their dissimilarity to real-world use cases of LMs (Raji et al., 2021), and in the means in which they were collected, for example by scraping (potentially copyrighted) material off of the internet in the case of MMLU. The use

of copyrighted material in the training and testing of AI systems is currently unsettled (Levendowski, 2021; Callison-Burch et al., 2023).

## Acknowledgements

We thank members of the Aristo team at AI2, Ari Holtzman, Peter West, Hanna Hajishirzi, and members of the H2 lab at the University of Washington for insightful feedback.

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

# A  Appendix

## A.1  Implementation Details

We use Huggingface Datasets (Lhoest et al., 2021) and Huggingface Transformers (Wolf et al., 2020) for implementation. All GPT-3 models were queried via the OpenAI API (https://beta.openai.com) between January and May 2023.

## A.2  Nature of Datasets for Each Model

For models trained only on the autoregressive next-token prediction objective (curie, davinci, and OPT 30B (Zhang et al., 2022)), in theory the OpenbookQA and CommonsenseQA datasets have not been seen in during training. However, guarantees would require access and indexing of the training corpora, which are not publicly available for the GPT-3 models. Additionally, due to the fact that training data was scraped for these models up to and including 2019 (Brown et al., 2020), it is possible there is some leakage in the training corpus.

For the instruction-tuned models, the authors of FLAN-T5 (Chung et al., 2022) explicitly report the datasets which are used and not used during training, and we report these details in §6.2. As for InstructGPT instruct-davinci-beta (Ouyang et al., 2022), the following details are given about its supervised instruction tuning training dataset (emphasis ours):

> "...The SFT dataset contains **about 13k training prompts (from the API and labeler-written)**...To give a sense of the composition of our dataset, in Table 1 we show the distribution of use-case categories for our API prompts (specifically the RM [reward modeling] dataset)

as labeled by our contractors. Most of the use-cases have (sp) are **generative, rather than classification or QA**. These prompts are **very diverse and include generation, question answering, dialog, summarization, extractions, and other natural language tasks** (see Table 1)."

In Table 1, generation makes up $45.6\%$ of the dataset, followed by open QA at $12.4\%$. Closed QA is a relatively small percentage of the training set, at $2.6\%$, and classification $3.5\%$, providing some possibility that the tasks we study are out-of-domain/zero-shot (though these exact numbers are reported on the reward modeling dataset, not the one used for instruction tuning, and these are not guarantees due to the proprietary nature of the dataset). No details are given about the datasets used to train text-davinci-003 (OpenAI, 2022).

## A.3  Prompt Details

Exemplar prompts containing 4 in-context demonstrations (for 1 of the 3 random seeds used) are given in Tables 2 to 4 for OpenbookQA and Tables 5 to 7 for CommonsenseQA. The last instance shown is the test instance, which the model completes with an answer prediction. For each random seed, 8 demonstrations are drawn from the training set of each dataset. When fewer demonstrations (0-4) are used, the first $k$ are taken and the prompt otherwise stays the same.

**Role of Different Parts of the Input** In Figs. 5 and 11 to 13, when prompts do not include $q$, we use the same prompts as in §6.3, minus the question.

For example, when $x = \mathcal{L}_{string}$:

```
answer choices: snacks, naps, kites, or warmth
The correct answer is:
```

When $x = \mathcal{L}_{enum}$:

```
Choices:
 A: snacks
 B: naps
 C: kites
 D: warmth
Answer:
```

When $x = None$, the prompt is simply "? " to avoid an empty context, which the OpenAI API does not allow.

## A.4  Computing $\mathrm{PMI_{DC}}$

In Holtzman et al. (2021), the denominator of Eq. (4) is actually computed as $P_\theta(\ell|d)$, where $d$

Bears will always have longer life cycles than a fox
If a river is rushing southwest on a sunny day, then it is safe to assume that the land gently inclines in that direction
After the moon phase where you can see nothing of the moon, what comes next? the first quarter
kinetics change stored energy into motion and warmth
A person wants to start saving money so that they can afford a nice vacation at the end of the year. After looking over their budget and expenses, they decide the best way to save money is to

Table 2: One of three "$q$" prompt templates used for OpenbookQA, containing 4 in-context demonstrations and one test instance.

Let's answer science questions.

question: Bears will always have longer life cycles than a
answer choices: tortoises, whales, elephants, or fox
The correct answer is: fox
###
question: If a river is rushing southwest on a sunny day, then it is safe to assume that
answer choices: southwest is a good place to be, the land gently inclines in that direction, the world is mostly land, or the land is supple
The correct answer is: the land gently inclines in that direction
###
question: After the moon phase where you can see nothing of the moon, what comes next?
answer choices: the full moon, the last quarter, the first quarter, or the half moon
The correct answer is: the first quarter
###
question: kinetics change stored energy into motion and
answer choices: snacks, naps, kites, or warmth
The correct answer is: warmth
###
question: A person wants to start saving money so that they can afford a nice vacation at the end of the year. After looking over their budget and expenses, they decide the best way to save money is to
answer choices: make more phone calls, quit eating lunch out, buy less with monopoly money, or have lunch with friends
The correct answer is:

Table 3: One of three "$q + \mathcal{L}_{string}$" prompt templates used for OpenbookQA, containing 4 in-context demonstrations and one test instance.

represents some "domain context" string. In their implementation, $d$ is the phrase " the answer is:". A context is necessary practically when querying the OpenAI API as well, as they do not allow queries with empty contexts, presumably to avoid revealing model weights. In our setting, we follow the prompt format to determine $d$. For $x = q$, $d =$ "? " (to avoid an empty context). Otherwise, $d$ is the last line of the prompt— for $x = q + \mathcal{L}_{string}$, $d =$ "The correct answer is: ", and for $x = q + \mathcal{L}_{enum}$, $d =$ "Answer: ".

Following Holtzman et al. (2021), $P_\theta(\ell|d)$ is always computed zero-shot, even when the numerator has in-context examples. We follow this design, as it is unintuitive to include in-context examples that do not contain a question, such as "The correct answer is: birds
The correct answer is: dogs
The correct answer is: ", and unclear how this would better calibrate a model's predictions.

We experimented with a "label-conditional" domain context where we included answer choices in $d$ when the prompt contained them, but found this version of $\text{PMI}_{\text{DC}}$, $\hat{y} = \frac{p(\ell|q+\mathcal{L})}{p(\ell|\mathcal{L})}$, to underperform the version without answer choices.

### A.5 Proofs

We say that a string $y$ forms a prefix of a string $y'$ if $y = y_1 \dots y_k$ and $y' = y_1 \dots y_k \dots y_m$ for $k < m$. We call a set $S$ of strings *prefix-free* if no string in $S$ is a prefix of another string in $S$. We show below (using two alternative arguments) that the total probability mass assigned by a language model $M_\theta$ to a prefix-free set is at most 1. Note that the prefix-free condition is necessary for the upper bound of 1 to hold in general.

**Proposition 1.** *For any prefix-free set $S$ of strings and any $x$, $\sum_{y \in S} P_\theta(y|x) \leq 1$.*

It follows that if the set $\mathcal{L}$ of answer choices is prefix-free, then $\text{PMA}_\theta(\mathcal{L}, x) \leq 1$.

The idea behind the first proof of Proposition 1 is that if multiple strings in $S$ share a common max-

Table 4: One of three "$q + \mathcal{L}_{enum}$" prompt templates used for OpenbookQA, containing 4 in-context demonstrations and one test instance.

imal prefix, the token that immediately follows that common prefix must be distinct across the strings (because $S$ is prefix-free). It follows from this that the *total* probability of those strings sharing the prefix is no more than the probability of the common prefix itself. We can use this observation to repeated *reduce* $S$ into strictly smaller sets that retain the invariant of being prefix-free and upper bound the total probability of the original $S$. The process end when no two strings share a common prefix, at which point, the upper bound of 1 follows immediately. Formally,

*Proof via maximal common prefixes.* Let $y_1 y_2 \ldots y_k$ denote the $k$ tokens comprising a string $y \in S$, where $k = |y|$ is the length of $y$.

Recall that $P_\theta(y|x) = \prod_{i=1}^{k} P_\theta(y_i|x, y_1 \ldots y_{i-1})$. We thus have $P_\theta(y|x) \leq P_\theta(y_1 \ldots y_l|x)$ for any $l \leq k$. In particular, $P_\theta(y|x) \leq P_\theta(y_1|x)$.

If it's the case that the first tokens of all strings $y \in S$ are distinct, then $\sum_{y \in S} P_\theta(y_1|x) \leq 1$ since $P_\theta$ is a probability distribution over tokens. It follows that $\sum_{y \in S} P_\theta(y|x) \leq 1$, finishing the proof.

If the first tokens are not all distinct, then there must exist at least two strings in $S$ that share a common prefix. We can therefore identify a maximal prefix $p$ and a subset $S' \subseteq S$ with $|S'| \geq 2$ such that all strings in $S'$ begin with the prefix $p$, while none of the ones in $S \setminus S'$ do. Since $p$ is a *maximal* prefix, the tokens $y_{|p|+1}$ of strings $y \in S'$ that immediately follow $p$ must all be distinct. Following

Fabric is cut to order at what type of seller? tailor shop
Where are you if your reading magazines while waiting for a vehicle on rails? train station
What would need oil to be used? combustion engines
What is person probably feeling that plans on stopping being married to their spouse? detachment
A revolving door is convenient for two direction travel, but it also serves as a security measure at a what?

Table 5: One of three "$q$" prompt templates used for CommonsenseQA, containing 4 in-context demonstrations and one test instance.

Let's answer commonsense reasoning questions.

question: Fabric is cut to order at what type of seller?
answer choices: hardware store, curtains, tailor shop, clothing store, or sewing room
The correct answer is: tailor shop
###
question: Where are you if your reading magazines while waiting for a vehicle on rails?
answer choices: bookstore, vegetables, market, doctor, or train station
The correct answer is: train station
###
question: What would need oil to be used?
answer choices: service station, ground, human body, repair shop, or combustion engines
The correct answer is: combustion engines
###
question: What is person probably feeling that plans on stopping being married to their spouse?
answer choices: wrong, detachment, bankruptcy, sad, or fights
The correct answer is: detachment
###
question: A revolving door is convenient for two direction travel, but it also serves as a security measure at a what?
answer choices: new york, bank, library, department store, or mall
The correct answer is:

Table 6: One of three "$q + \mathcal{L}_{string}$" prompt templates used for CommonsenseQA, containing 4 in-context demonstrations and one test instance.

the argument used earlier, we have:

$$\sum_{y \in S'} P_\theta(y|x) \leq \sum_{y \in S'} P_\theta(y_1 \ldots y_{|p|+1}|x)$$
$$= \sum_{y \in S'} P_\theta(py_{|p|+1}|x)$$
$$= \sum_{y \in S'} P_\theta(p|x)P_\theta(y_{|p|+1}|xp)$$
$$= P_\theta(p|x) \sum_{y \in S'} P_\theta(y_{|p|+1}|xp)$$
$$\leq P_\theta(p|x)$$

where the last inequality holds because $P_\theta$ is a probability distribution and the tokens $y_{|p|+1}$ are all distinct as observed above. Now consider the set $T = (S \setminus S') \cup \{p\}$, i.e., the original set $S$ except with all strings beginning with the prefix $p$ replaced

with a single string $p$. Observe that:

$$\sum_{y \in S} P_\theta(y|x) = \sum_{y \in S \setminus S'} P_\theta(y|x) + \sum_{y \in S'} P_\theta(y|x)$$
$$\leq \Big( \sum_{y \in S \setminus S'} P_\theta(y|x) \Big) + P_\theta(p|x)$$
$$= \sum_{y \in T} P_\theta(y|x).$$

That is, the total probability mass over strings in $S$ is upper bounded by that on strings in $T$. Further, $T$ contains $p$ and has size $|S| - |S'| + 1$, and we know $|S'| \geq 2$. Thus, $1 \leq |T| < |S|$. If $|T| = 1$, we immediately have $\sum_{y \in T} P_\theta(y|x) \leq 1$ and the proof is complete. Otherwise, we observe that $T$ is also prefix-free just like $S$, so we can simplify $S$ to be $T$ and repeat the process of checking the distinctness of first tokens, identifying the maximal prefix, and further upper bounding the probability mass on $S$. Since each iteration of this process reduces the size of $S$ by at least 1, the process must terminate with $|S| = 1$, at which point we conclude that the total probability mass on the reduced $S$—and hence on the original $S$—is at most 1, as claimed. $\square$

An alternative argument for proving Proposition 1 is to *expand* the set $S$ into a larger set $T$ such that (a) the total probability mass on $S$ is the same as that on $T$ and (b) all strings in $T$ are of the same length, say $k$. We can then observe that the total probability mass on $T$ is upper bounded by the probability mass on *all* strings of length $k$, and argue that the latter is exactly 1.

*Proof using length normalization.* Let $k$ denote the length of the longest string in $S$ and $\mathcal{V}$ denote the token vocabulary. For any string $y$ of length at most $k$, let $Z_y^k$ denote the set of *all* possible extensions of $y$ to strings of length exactly $k$. We first argue that $P_\theta(y|x) = \sum_{y' \in Z_y^k} P_\theta(y'|x)$:

$$\sum_{y' \in Z_y^k} P_\theta(y'|x)$$
$$= \sum_{y'_{|y|+1} \in \mathcal{V}} \cdots \sum_{y'_k \in \mathcal{V}} P_\theta(yy'_{|y|+1} \ldots y'_k|x)$$
$$= P_\theta(y|x) \sum_{y'_{|y|+1} \in \mathcal{V}} \cdots \sum_{y'_k \in \mathcal{V}} P_\theta(y'_{|y|+1}|xy) \times \ldots$$
$$\times P_\theta(y'_k|xy \ldots y'_{k-1})$$
$$= P_\theta(y|x) \left( \sum_{y'_{|y|+1} \in \mathcal{V}} P_\theta(y'_{|y|+1}|xy) \right) \times \ldots$$
$$\times \left( \sum_{y'_k \in \mathcal{V}} \ldots P_\theta(y'_k|xy \ldots y'_{k-1}) \right)$$
$$= P_\theta(y|x) \times 1 \times \ldots \times 1$$
$$= P_\theta(y|x)$$

where the equality with 1 in the second-last line follows because $P_\theta$ is a probability distribution over tokens in $\mathcal{V}$.

Now define an expanded set $T$ as the set of all expansions of strings in $S$ to strings of lengths exactly $k$. Since $S$ is prefix-free, all of these expanded strings are distinct. Thus, it follows from the above argument that the total probability mass on $S$ is the same as that on $T$. Lastly, the probability mass on $T$ is clearly upper bounded by that on *all* length-$k$ strings, which itself is exactly 1 (using the same argument as in the second-last line of the derivation above). Therefore, the total probability mass on $S$ is upper bounded by 1. □

**Empirical Verification of Prefix-Free Assumption** Empirically,[12] only 24/1140 MMLU instances contain an answer choice that is a prefix of another answer choice in $\mathcal{L}$, 5/500 in CommonsenseQA, and 0/500 in OpenbookQA. In the case of MMLU, 24 instances is an upper bound since many of the prefix answers are numeric; whether the answer choices are true prefixes would depend on the model's tokenizer (e.g., whether "2" and "200" have the same first token after tokenization will vary).

### A.6 Additional Results

- Table 8 contains the tables of bound satisfaction (Eq. (7)) for all datasets.

- Table 9 contains tabular results for MMLU; Table 10 for CommonsenseQA and Table 11 for OpenbookQA.

- Fig. 7 contains scatterplots for CommonsenseQA and OpenbookQA.

- Fig. 8 contains $\text{PMI}_{\text{DC}}$ vs. sequence-scoring accuracy heatmaps for OpenbookQA and CommonsenseQA.

- Figs. 9 and 10 contain line graphs for CommonsenseQA and OpenbookQA, respectively.

- Figs. 11 to 13 show barcharts for accuracy and probability mass on given answer choices conditioned on various combinations of independent variables in the prompt, for all 3 datasets.

---

[12]After removing instances where two answer choices are duplicates, which is an artifact of dataset collection that can easily be resolved (1 instance in MMLU, 10 in CommonsenseQA, and 0 in OpenbookQA).

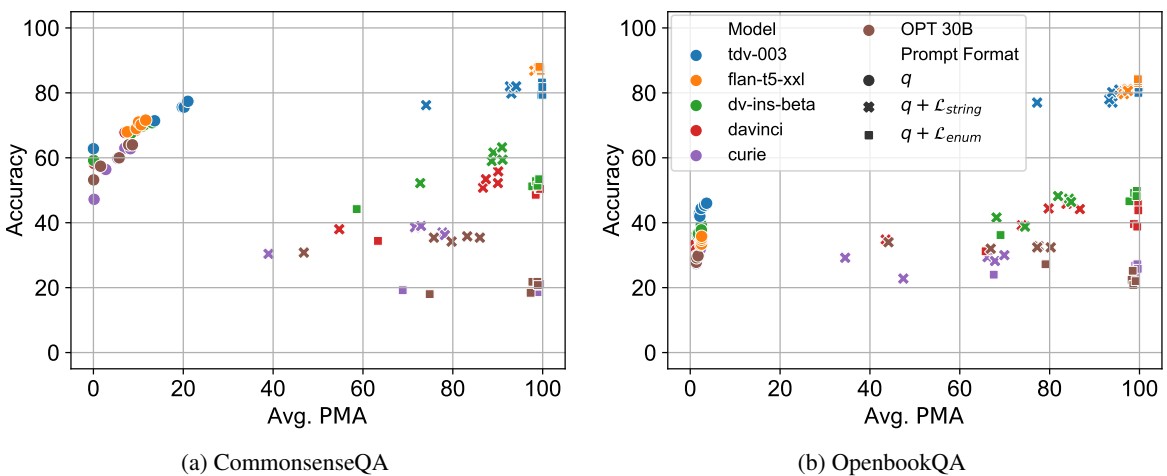

(a) CommonsenseQA

(b) OpenbookQA

Figure 7: A scatterplot showing the relationship between average PMA and task accuracy for 0, 1, 2, 4 and 8 in-context examples. Note these datasets are explicitly in-domain for FLAN-T5. See Figure 4 for more info.

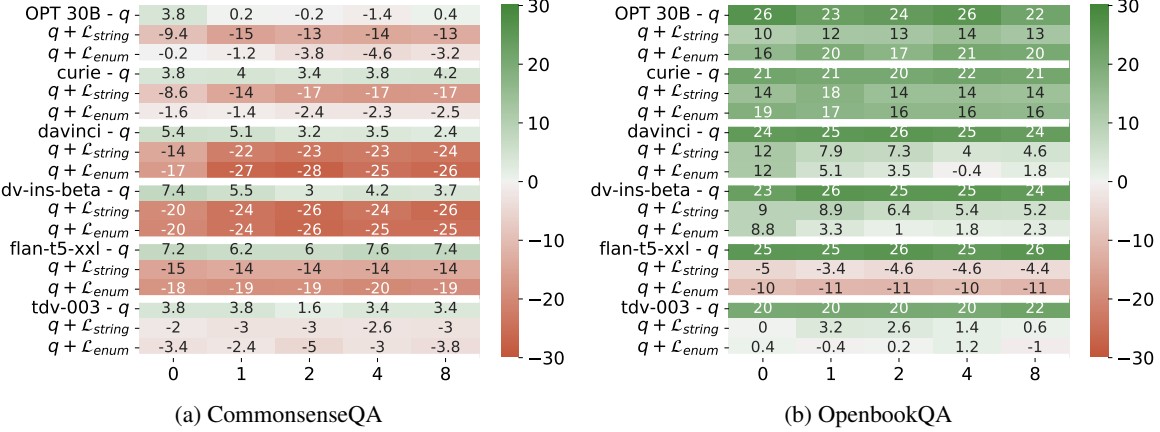

(a) CommonsenseQA

(b) OpenbookQA

Figure 8: Accuracy changes achieved by using $\text{PMI}_{\text{DC}}$ (Eq. (4)) over standard sequence scoring (Eq. (1)). Full accuracy scores are in Tables 9 to 11.

The following are multiple-choice questions about everyday situations. For the question below, select the most suitable answer from the 5 options given.

Question: Fabric is cut to order at what type of seller?
Choices:
   A: curtains
   B: tailor shop
   C: clothing store
   D: sewing room
   E: hardware store
Answer: B

Question: Where are you if your reading magazines while waiting for a vehicle on rails?
Choices:
   A: vegetables
   B: market
   C: doctor
   D: train station
   E: bookstore
Answer: D

Question: What would need oil to be used?
Choices:
   A: ground
   B: human body
   C: repair shop
   D: combustion engines
   E: service station
Answer: D

Question: What is person probably feeling that plans on stopping being married to their spouse?
Choices:
   A: detachment
   B: bankruptcy
   C: sad
   D: fights
   E: wrong
Answer: A

Question: A revolving door is convenient for two direction travel, but it also serves as a security measure at a what?
Choices:
   A: bank
   B: library
   C: department store
   D: mall
   E: new york
Answer:

Table 7: One of three "$q + \mathcal{L}_{enum}$" prompt templates used for CommonsenseQA, containing 4 in-context demonstrations and one test instance.

| Model Name | Prompt Format | Dataset | # In-Context Demonstrations | | | | |
|---|---|---|---|---|---|---|---|
| | | | 0 | 1 | 2 | 4 | 8 |
| OPT 30B | $q$ | MMLU | 0.18 | 0.88 | 0.61 | 1.05 | 2.19 |
| | | CommonsenseQA | 0.0 | 0.0 | 1.2 | 2.6 | 3.6 |
| | | OpenbookQA | 0.6 | 0.4 | 0.6 | 0.6 | 0.4 |
| | $q + \mathcal{L}_{string}$ | MMLU | 6.23 | 37.54 | 40.53 | 45.26 | 49.21 |
| | | CommonsenseQA | 2.4 | 29.4 | 38.2 | 47.2 | 52.2 |
| | | OpenbookQA | 4.8 | 25.8 | 42.4 | 45.2 | 48.6 |
| | $q + \mathcal{L}_{enum}$ | MMLU | 3.68 | 78.25 | 78.25 | 82.81 | 88.6 |
| | | CommonsenseQA | 0.0 | 54.6 | 63.4 | 74.6 | 78.2 |
| | | OpenbookQA | 0.2 | 65.2 | 78.2 | 78.0 | 85.4 |
| GPT-3 curie (~6.7B) | $q$ | MMLU | 0.35 | $1.37_{0.56}$ | $1.2_{0.94}$ | $0.94_{0.56}$ | $0.91_{0.13}$ |
| | | CommonsenseQA | 0.0 | $0.07_{0.12}$ | $0.6_{0.53}$ | $2.13_{0.5}$ | $3.8_{0.4}$ |
| | | OpenbookQA | 0.4 | $0.4_{0.0}$ | $0.4_{0.0}$ | $0.4_{0.0}$ | $0.47_{0.12}$ |
| | $q + \mathcal{L}_{string}$ | MMLU | 3.86 | $25.11_{1.71}$ | $30.12_{0.9}$ | $38.01_{1.58}$ | $44.53_{0.68}$ |
| | | CommonsenseQA | 1.8 | $28.27_{1.21}$ | $32.0_{0.6}$ | $37.87_{2.81}$ | $40.8_{4.13}$ |
| | | OpenbookQA | 1.2 | $21.6_{10.89}$ | $35.07_{12.7}$ | $36.6_{11.48}$ | $32.8_{6.92}$ |
| | $q + \mathcal{L}_{enum}$ | MMLU | 0.0 | $78.83_{2.26}$ | $86.05_{0.79}$ | $91.08_{0.82}$ | $93.62_{0.99}$ |
| | | CommonsenseQA | 0.0 | $71.87_{11.7}$ | $82.87_{11.96}$ | $98.27_{0.12}$ | $98.0_{0.53}$ |
| | | OpenbookQA | 0.0 | $76.87_{10.85}$ | $84.53_{11.27}$ | $91.27_{4.96}$ | $91.67_{1.17}$ |
| GPT-3 davinci (~175B) | $q$ | MMLU | 0.53 | $2.75_{0.28}$ | $2.49_{0.98}$ | $3.33_{0.46}$ | $4.62_{0.4}$ |
| | | CommonsenseQA | 0.0 | $1.8_{1.71}$ | $4.53_{0.31}$ | $6.13_{0.58}$ | $7.33_{1.3}$ |
| | | OpenbookQA | 0.2 | $0.73_{0.31}$ | $0.4_{0.2}$ | $0.4_{0.35}$ | $0.6_{0.2}$ |
| | $q + \mathcal{L}_{string}$ | MMLU | 2.89 | $51.87_{1.4}$ | $59.07_{1.7}$ | $64.06_{1.72}$ | $66.81_{1.25}$ |
| | | CommonsenseQA | 6.8 | $61.8_{6.12}$ | $62.33_{14.87}$ | $67.53_{12.08}$ | $74.4_{6.97}$ |
| | | OpenbookQA | 4.0 | $48.47_{13.27}$ | $67.73_{12.91}$ | $61.73_{11.71}$ | $64.07_{6.74}$ |
| | $q + \mathcal{L}_{enum}$ | MMLU | 1.14 | $88.19_{1.63}$ | $94.21_{0.09}$ | $96.7_{0.73}$ | $98.36_{0.42}$ |
| | | CommonsenseQA | 0.0 | $87.67_{3.14}$ | $96.2_{0.2}$ | $97.8_{0.2}$ | $98.4_{0.4}$ |
| | | OpenbookQA | 0.0 | $92.8_{1.59}$ | $96.73_{0.7}$ | $97.8_{0.53}$ | $99.0_{0.53}$ |
| davinci-instruct-beta | $q$ | MMLU | 0.88 | $1.34_{0.53}$ | $2.1_{0.85}$ | $2.69_{0.05}$ | $4.24_{0.37}$ |
| | | CommonsenseQA | 0.0 | $3.07_{1.36}$ | $5.4_{0.6}$ | $6.73_{0.42}$ | $7.0_{0.87}$ |
| | | OpenbookQA | 0.4 | $0.87_{0.31}$ | $0.67_{0.23}$ | $0.6_{0.2}$ | $0.73_{0.12}$ |
| | $q + \mathcal{L}_{string}$ | MMLU | 32.72 | $65.56_{1.77}$ | $68.89_{0.98}$ | $70.85_{0.57}$ | $72.75_{0.05}$ |
| | | CommonsenseQA | 46.8 | $69.07_{7.16}$ | $76.0_{4.57}$ | $79.6_{3.34}$ | $80.8_{4.2}$ |
| | | OpenbookQA | 36.4 | $60.93_{16.15}$ | $70.13_{10.72}$ | $68.87_{9.2}$ | $72.0_{5.96}$ |
| | $q + \mathcal{L}_{enum}$ | MMLU | 20.79 | $93.71_{0.36}$ | $95.94_{0.34}$ | $96.35_{0.68}$ | $97.92_{0.86}$ |
| | | CommonsenseQA | 15.6 | $94.8_{0.35}$ | $97.2_{0.35}$ | $98.0_{0.4}$ | $98.13_{0.42}$ |
| | | OpenbookQA | 33.4 | $95.8_{2.03}$ | $98.2_{0.35}$ | $99.0_{0.53}$ | $99.0_{0.4}$ |
| FLAN-T5-XXL (11B) | $q$ | MMLU | 0.79 | 1.05 | 1.75 | 1.58 | 2.28 |
| | | CommonsenseQA | 3.4 | 5.0 | 6.0 | 6.0 | 7.8 |
| | | OpenbookQA | 0.6 | 0.8 | 0.6 | 0.6 | 0.6 |
| | $q + \mathcal{L}_{string}$ | MMLU | 85.96 | 88.86 | 89.74 | 90.18 | 90.79 |
| | | CommonsenseQA | 98.0 | 99.4 | 99.4 | 99.0 | 98.6 |
| | | OpenbookQA | 95.6 | 95.2 | 96.4 | 97.4 | 96.8 |
| | $q + \mathcal{L}_{enum}$ | MMLU | 98.86 | 98.33 | 98.68 | 98.68 | 98.33 |
| | | CommonsenseQA | 99.2 | 99.2 | 99.4 | 99.4 | 99.0 |
| | | OpenbookQA | 99.8 | 100.0 | 100.0 | 99.6 | 100.0 |
| text-davinci-003 | $q$ | MMLU | 0.88 | 3.77 | 7.11 | 9.47 | 10.61 |
| | | CommonsenseQA | 0.0 | 11.6 | 19.6 | 18.8 | 19.6 |
| | | OpenbookQA | 1.6 | 1.6 | 1.8 | 2.6 | 3.0 |
| | $q + \mathcal{L}_{string}$ | MMLU | 91.75 | 94.56 | 95.96 | 96.67 | 96.93 |
| | | CommonsenseQA | 80.4 | 92.8 | 91.8 | 93.2 | 93.4 |
| | | OpenbookQA | 82.0 | 94.4 | 93.6 | 94.0 | 95.2 |
| | $q + \mathcal{L}_{enum}$ | MMLU | 99.91 | 100.0 | 99.91 | 100.0 | 99.82 |
| | | CommonsenseQA | 99.8 | 100.0 | 100.0 | 99.8 | 100.0 |
| | | OpenbookQA | 99.8 | 100.0 | 99.8 | 99.8 | 100.0 |

Table 8: % of instances for which Eq. (7) is true and thus SFC could not have affected the model's prediction, shown here for all combinations of models, datasets, and prompt formats considered.

| Model Name | Prompt Format | Metric | 0 | 1 | 2 | 4 | 8 |
|---|---|---|---|---|---|---|---|
| | | | **# In-Context Demonstrations** | | | | |
| OPT 30B | $q$ | Accuracy | 32.98 | **33.86** | 34.74 | 34.56 | 35.26 |
| | | PMI$_{DC}$ Acc. | **35.18** | **34.65** | **35.96** | **36.84** | **36.84** |
| | | Avg. PMA | 0.74 | 2.53 | 3.35 | 5.12 | 5.93 |
| | $q + \mathcal{L}_{string}$ | Accuracy | 31.49 | 30.79 | 30.61 | 32.19 | 30.7 |
| | | PMI$_{DC}$ Acc. | 27.72 | 28.51 | 27.98 | 27.98 | 28.33 |
| | | Avg. PMA | 44.67 | 71.43 | 73.7 | 75.61 | 76.74 |
| | $q + \mathcal{L}_{enum}$ | Accuracy | 26.23 | 27.28 | 26.84 | 25.61 | 25.88 |
| | | PMI$_{DC}$ Acc. | 28.25 | 27.81 | 28.16 | 28.16 | 28.51 |
| | | Avg. PMA | 81.04 | 97.94 | 98.5 | 98.86 | 99.04 |
| GPT-3 curie (~6.7B) | $q$ | Accuracy | 34.04 | $34.15_{0.58}$ | $34.71_{1.04}$ | $35.29_{0.53}$ | $36.29_{0.31}$ |
| | | PMI$_{DC}$ Acc. | **36.4** | $\mathbf{36.87_{1.23}}$ | $\mathbf{38.36_{0.27}}$ | $\mathbf{39.24_{0.45}}$ | $\mathbf{39.15_{0.84}}$ |
| | | Avg. PMA | 0.82 | $2.64_{0.17}$ | $3.55_{0.19}$ | $4.28_{0.09}$ | $5.16_{0.27}$ |
| | $q + \mathcal{L}_{string}$ | Accuracy | 30.26 | $31.2_{1.44}$ | $29.85_{0.58}$ | $31.11_{0.96}$ | $31.72_{1.59}$ |
| | | PMI$_{DC}$ Acc. | 31.32 | $31.02_{0.48}$ | $30.58_{0.31}$ | $30.41_{0.35}$ | $30.82_{0.7}$ |
| | | Avg. PMA | 32.77 | $57.94_{0.34}$ | $65.05_{0.24}$ | $71.4_{0.41}$ | $75.74_{0.58}$ |
| | $q + \mathcal{L}_{enum}$ | Accuracy | 22.81 | $26.37_{0.13}$ | $26.52_{0.89}$ | $25.32_{1.49}$ | $27.1_{0.83}$ |
| | | PMI$_{DC}$ Acc. | 29.65 | $29.65_{0.23}$ | $29.39_{0.35}$ | $29.33_{0.18}$ | $29.15_{0.28}$ |
| | | Avg. PMA | 69.61 | $98.54_{0.02}$ | $98.94_{0.01}$ | $99.23_{0.01}$ | $99.46_{0.01}$ |
| GPT-3 davinci (~175B) | $q$ | Accuracy | $37.81$ | $38.51_{0.23}$ | $39.73_{0.57}$ | $40.32_{0.5}$ | $40.96_{0.83}$ |
| | | PMI$_{DC}$ Acc. | **41.32** | $\mathbf{41.84_{0.85}}$ | $\mathbf{42.69_{0.33}}$ | $\mathbf{43.83_{0.89}}$ | $\mathbf{44.59_{0.31}}$ |
| | | Avg. PMA | 1.18 | $4.35_{0.4}$ | $5.84_{0.11}$ | $7.06_{0.14}$ | $7.98_{0.28}$ |
| | $q + \mathcal{L}_{string}$ | Accuracy | 34.74 | $38.07_{0.84}$ | $39.74_{0.75}$ | $39.77_{0.73}$ | $40.26_{0.27}$ |
| | | PMI$_{DC}$ Acc. | 30.53 | $31.58_{0.61}$ | $31.99_{0.45}$ | $32.02_{0.3}$ | $32.51_{0.35}$ |
| | | Avg. PMA | 39.0 | $80.14_{0.51}$ | $83.48_{0.15}$ | $85.82_{0.04}$ | $87.03_{0.03}$ |
| | $q + \mathcal{L}_{enum}$ | Accuracy | 36.49 | $\mathbf{41.23_{1.29}}$ | $\mathbf{42.81_{0.09}}$ | $\mathbf{43.51_{0.35}}$ | $\mathbf{44.12_{0.69}}$ |
| | | PMI$_{DC}$ Acc. | 27.54 | $28.83_{0.28}$ | $29.36_{0.18}$ | $29.94_{0.49}$ | $30.0_{0.31}$ |
| | | Avg. PMA | 67.54 | $98.7_{0.06}$ | $99.2_{0.02}$ | $99.51_{0.02}$ | $99.7_{0.01}$ |
| davinci-instruct-beta | $q$ | Accuracy | 37.37 | $38.07_{0.4}$ | $39.15_{0.54}$ | $40.44_{0.81}$ | $40.91_{0.51}$ |
| | | PMI$_{DC}$ Acc. | **41.32** | $\mathbf{42.14_{0.89}}$ | $\mathbf{43.63_{0.53}}$ | $\mathbf{44.3_{0.61}}$ | $\mathbf{44.33_{0.75}}$ |
| | | Avg. PMA | 1.1 | $3.42_{0.43}$ | $5.05_{0.24}$ | $6.7_{0.22}$ | $7.68_{0.28}$ |
| | $q + \mathcal{L}_{string}$ | Accuracy | 37.02 | $38.62_{0.85}$ | $39.24_{2.06}$ | $40.38_{0.67}$ | $40.11_{1.17}$ |
| | | PMI$_{DC}$ Acc. | 31.49 | $31.96_{0.48}$ | $32.43_{0.48}$ | $32.25_{0.48}$ | $33.13_{0.31}$ |
| | | Avg. PMA | 68.54 | $85.16_{0.02}$ | $86.36_{0.18}$ | $87.81_{0.15}$ | $88.73_{0.15}$ |
| | $q + \mathcal{L}_{enum}$ | Accuracy | 40.0 | $40.12_{1.25}$ | $41.49_{0.7}$ | $42.31_{0.25}$ | $42.83_{0.1}$ |
| | | PMI$_{DC}$ Acc. | 29.39 | $30.09_{0.35}$ | $30.03_{0.14}$ | $30.23_{0.35}$ | $29.68_{0.18}$ |
| | | Avg. PMA | 67.55 | $97.64_{0.1}$ | $98.52_{0.09}$ | $98.72_{0.05}$ | $99.19_{0.05}$ |
| FLAN-T5-XXL (11B) | $q$ | Accuracy | 34.04 | 34.21 | 34.21 | 33.6 | 34.91 |
| | | PMI$_{DC}$ Acc. | 39.04 | 39.91 | 38.86 | 39.65 | 40.44 |
| | | Avg. PMA | 2.68 | 3.41 | 4.24 | 4.62 | 4.77 |
| | $q + \mathcal{L}_{string}$ | Accuracy | 51.75 | 52.89 | 53.68 | 53.51 | 53.68 |
| | | PMI$_{DC}$ Acc. | 39.12 | 39.82 | 39.91 | 39.91 | 40.0 |
| | | Avg. PMA | 91.84 | 92.89 | 93.37 | 94.0 | 94.54 |
| | $q + \mathcal{L}_{enum}$ | Accuracy | **53.86** | **54.3** | **55.18** | **55.35** | **55.79** |
| | | PMI$_{DC}$ Acc. | 37.37 | 38.25 | 39.21 | 39.21 | 39.65 |
| | | Avg. PMA | 99.18 | 99.39 | 99.42 | 99.38 | 99.36 |
| text-davinci-003 | $q$ | Accuracy | 44.91 | 47.54 | 49.65 | 50.88 | 52.11 |
| | | PMI$_{DC}$ Acc. | 50.7 | 51.93 | 54.21 | 56.49 | 57.19 |
| | | Avg. PMA | 1.23 | 4.87 | 8.43 | 11.03 | 12.36 |
| | $q + \mathcal{L}_{string}$ | Accuracy | 60.61 | **64.39** | 65.18 | **66.32** | **67.81** |
| | | PMI$_{DC}$ Acc. | 52.46 | 55.18 | 55.7 | 55.96 | 55.0 |
| | | Avg. PMA | 90.95 | 94.98 | 96.63 | 97.25 | 97.82 |
| | $q + \mathcal{L}_{enum}$ | Accuracy | **63.77** | **65.26** | **66.67** | **65.96** | **67.37** |
| | | PMI$_{DC}$ Acc. | 54.04 | 52.89 | 53.42 | 53.95 | 53.16 |
| | | Avg. PMA | 99.77 | 99.9 | 99.86 | 99.85 | 99.82 |

Table 9: Full metrics for each model and prompt type on the MMLU test subset. Models are ordered by increasing performance. The mean and standard error of using 3 random seeds to select in-context demonstrations are reported for experiments with at least 1 demonstration for the curie, davinci, and davinci-instruct-beta models. For each model and each column, we **bold** the prompt format and scoring metric (accuracy or PMI$_{DC}$ accuracy) that results in the highest score, as well as any scores within 1 percentage point of it. We underline the prompt format with the largest average PMA.

| Model Name | Prompt Format | Metric | # In-Context Demonstrations | | | | |
|---|---|---|---|---|---|---|---|
| | | | 0 | 1 | 2 | 4 | 8 |
| OPT 30B | $q$ | Accuracy | 53.2 | **57.4** | **60.0** | **64.0** | **64.0** |
| | | PMI$_{DC}$ Acc. | **57.0** | **57.6** | **59.8** | 62.6 | **64.4** |
| | | Avg. PMA | 0.05 | 1.59 | 5.74 | 7.84 | 8.68 |
| | $q + \mathcal{L}_{string}$ | Accuracy | 30.8 | 35.4 | 34.2 | 35.8 | 35.4 |
| | | PMI$_{DC}$ Acc. | 21.4 | 20.2 | 20.8 | 21.6 | 22.0 |
| | | Avg. PMA | 46.84 | 75.78 | 79.73 | 83.2 | 86.02 |
| | $q + \mathcal{L}_{enum}$ | Accuracy | 18.0 | 18.4 | 21.8 | 21.8 | 20.8 |
| | | PMI$_{DC}$ Acc. | 17.8 | 17.2 | 18.0 | 17.2 | 17.6 |
| | | Avg. PMA | 74.85 | 97.31 | 97.71 | 98.81 | 98.92 |
| GPT-3 curie (~6.7B) | $q$ | Accuracy | 47.2(40.0) | $52.4_{3.82}$ | $58.33_{1.5}$ | $61.4_{1.83}$(52.3) | $62.33_{1.36}$ |
| | | PMI$_{DC}$ Acc. | **51.0**(50.3) | $\mathbf{56.4_{2.91}}$ | $\mathbf{61.73_{1.3}}$ | $\mathbf{65.2_{1.64}}$(56.5) | $\mathbf{66.53_{1.3}}$ |
| | | Avg. PMA | 0.15 | $1.69_{0.92}$ | $4.03_{1.23}$ | $6.88_{0.98}$ | $8.43_{0.51}$ |
| | $q + \mathcal{L}_{string}$ | Accuracy | 30.4 | $37.6_{3.8}$ | $39.4_{2.03}$ | $40.2_{4.39}$ | $40.07_{4.63}$ |
| | | PMI$_{DC}$ Acc. | 21.8 | $23.2_{0.53}$ | $22.87_{0.61}$ | $22.93_{0.61}$ | $23.13_{1.22}$ |
| | | Avg. PMA | 38.98 | $71.26_{0.92}$ | $74.04_{1.03}$ | $77.04_{0.68}$ | $79.26_{1.08}$ |
| | $q + \mathcal{L}_{enum}$ | Accuracy | 19.2 | $19.6_{1.91}$ | $21.0_{0.4}$ | $21.27_{0.12}$ | $21.2_{0.0}$ |
| | | PMI$_{DC}$ Acc. | 17.6 | $18.2_{0.8}$ | $18.6_{0.35}$ | $19.0_{0.72}$ | $18.73_{0.9}$ |
| | | Avg. PMA | 68.84 | $98.74_{0.17}$ | $98.99_{0.07}$ | $99.23_{0.07}$ | $99.3_{0.05}$ |
| GPT-3 davinci (~175B) | $q$ | Accuracy | 58.2(61.0) | $63.47_{5.13}$ | $68.93_{1.29}$ | $70.53_{0.5}$(69.1) | $71.33_{1.27}$ |
| | | PMI$_{DC}$ Acc. | **63.6**(66.7) | $\mathbf{68.6_{3.61}}$ | $\mathbf{72.13_{1.62}}$ | $\mathbf{74.0_{1.91}}$(72.0) | $\mathbf{73.73_{0.81}}$ |
| | | Avg. PMA | 0.19 | $4.61_{2.48}$ | $8.85_{0.55}$ | $11.12_{0.82}$ | $12.02_{0.65}$ |
| | $q + \mathcal{L}_{string}$ | Accuracy | 38.0 | $51.27_{0.5}$ | $55.2_{3.83}$ | $56.33_{3.92}$ | $57.27_{1.5}$ |
| | | PMI$_{DC}$ Acc. | 23.8 | $29.2_{0.53}$ | $31.73_{1.79}$ | $33.73_{2.89}$ | $33.67_{1.22}$ |
| | | Avg. PMA | 54.72 | $85.62_{2.33}$ | $82.71_{9.12}$ | $85.33_{6.97}$ | $87.56_{3.77}$ |
| | $q + \mathcal{L}_{enum}$ | Accuracy | 34.4 | $48.27_{1.72}$ | $50.8_{0.87}$ | $49.6_{2.31}$ | $53.67_{3.61}$ |
| | | PMI$_{DC}$ Acc. | 17.8 | $21.73_{0.61}$ | $23.13_{1.01}$ | $25.07_{1.14}$ | $27.47_{0.46}$ |
| | | Avg. PMA | 63.34 | $98.46_{0.33}$ | $99.15_{0.07}$ | $99.37_{0.06}$ | $99.36_{0.07}$ |
| davinci-instruct-beta | $q$ | Accuracy | 59.2 | $65.27_{3.37}$ | $69.0_{1.39}$ | $69.8_{0.53}$ | $70.6_{0.92}$ |
| | | PMI$_{DC}$ Acc. | **66.6** | $\mathbf{70.73_{2.23}}$ | $\mathbf{72.0_{1.06}}$ | $\mathbf{74.0_{1.04}}$ | $\mathbf{74.27_{0.7}}$ |
| | | Avg. PMA | 0.05 | $5.55_{2.68}$ | $9.61_{1.27}$ | $11.45_{0.84}$ | $12.36_{0.65}$ |
| | $q + \mathcal{L}_{string}$ | Accuracy | 52.2 | $60.6_{3.3}$ | $63.73_{2.01}$ | $61.47_{2.47}$ | $62.13_{1.22}$ |
| | | PMI$_{DC}$ Acc. | 32.4 | $36.6_{1.4}$ | $38.07_{1.29}$ | $37.6_{1.78}$ | $36.33_{0.81}$ |
| | | Avg. PMA | 72.75 | $83.87_{6.35}$ | $87.32_{3.47}$ | $88.94_{3.06}$ | $89.73_{1.96}$ |
| | $q + \mathcal{L}_{enum}$ | Accuracy | 44.2 | $51.33_{1.8}$ | $53.2_{0.53}$ | $53.47_{2.47}$ | $54.93_{2.32}$ |
| | | PMI$_{DC}$ Acc. | 24.0 | $27.27_{0.81}$ | $26.73_{0.61}$ | $28.47_{1.6}$ | $30.0_{1.06}$ |
| | | Avg. PMA | 58.6 | $97.66_{0.43}$ | $98.62_{0.14}$ | $98.75_{0.11}$ | $99.06_{0.19}$ |
| FLAN-T5-XXL (11B) | $q$ | Accuracy | 68.0 | 69.0 | 71.0 | 70.2 | 71.6 |
| | | PMI$_{DC}$ Acc. | 75.2 | 75.2 | 77.0 | 77.8 | 79.0 |
| | | Avg. PMA | 7.54 | 9.54 | 10.05 | 10.62 | 11.64 |
| | $q + \mathcal{L}_{string}$ | Accuracy | **86.8** | **87.8** | **88.0** | **87.6** | **87.6** |
| | | PMI$_{DC}$ Acc. | 71.8 | 74.0 | 74.2 | 73.6 | 74.0 |
| | | Avg. PMA | 98.16 | 98.95 | 99.11 | 99.15 | 99.15 |
| | $q + \mathcal{L}_{enum}$ | Accuracy | **87.2** | 86.8 | 86.8 | **87.6** | **88.0** |
| | | PMI$_{DC}$ Acc. | 69.2 | 68.0 | 68.2 | 67.6 | 68.8 |
| | | Avg. PMA | 99.48 | 99.53 | 99.56 | 99.49 | 99.27 |
| text-davinci-003 | $q$ | Accuracy | 62.8 | 71.4 | 75.6 | 75.6 | 77.4 |
| | | PMI$_{DC}$ Acc. | 66.6 | 75.2 | 77.2 | 79.0 | 80.8 |
| | | Avg. PMA | 0.0 | 13.62 | 19.78 | 20.19 | 21.03 |
| | $q + \mathcal{L}_{string}$ | Accuracy | 76.2 | **79.8** | **82.0** | 81.6 | **82.0** |
| | | PMI$_{DC}$ Acc. | 74.2 | 76.8 | 79.0 | 79.0 | 79.0 |
| | | Avg. PMA | 74.06 | 93.04 | 92.71 | 93.68 | 94.1 |
| | $q + \mathcal{L}_{enum}$ | Accuracy | **79.4** | **79.4** | **82.8** | **83.2** | **81.8** |
| | | PMI$_{DC}$ Acc. | 76.0 | 77.0 | 77.8 | 80.2 | 78.0 |
| | | Avg. PMA | 99.61 | 99.95 | 99.95 | 99.85 | 99.92 |

Table 10: Full metrics for each model and prompt type on the CommonsenseQA validation subset. See caption of Table 9 for more details. #s in parentheses are those reported in Holtzman et al. (2021), though exact model used may not be the same.

| Model Name | Prompt Format | Metric | # In-Context Demonstrations | | | | |
|---|---|---|---|---|---|---|---|
| | | | 0 | 1 | 2 | 4 | 8 |
| OPT 30B | $q$ | Accuracy | 29.2 | 27.8 | 29.2 | 29.8 | 34.2 |
| | | $PMI_{DC}$ Acc. | **55.6** | **51.0** | **53.2** | **55.4** | **55.8** |
| | | Avg. PMA | 1.29 | 1.34 | 1.43 | 1.74 | 2.27 |
| | $q + \mathcal{L}_{string}$ | Accuracy | 34.0 | 32.0 | 32.8 | 32.4 | 32.4 |
| | | $PMI_{DC}$ Acc. | 44.4 | 44.4 | 45.4 | 46.6 | 45.0 |
| | | Avg. PMA | 44.13 | 66.89 | 77.52 | 77.23 | 80.25 |
| | $q + \mathcal{L}_{enum}$ | Accuracy | 27.2 | 22.4 | 25.2 | 20.8 | 22.0 |
| | | $PMI_{DC}$ Acc. | 43.0 | 42.0 | 42.2 | 41.6 | 42.2 |
| | | Avg. PMA | 79.08 | 98.24 | 98.49 | 98.63 | 99.14 |
| GPT-3 curie (~6.7B) | $q$ | Accuracy | 29.0(22.4) | $28.67_{1.17}$ | $29.53_{2.2}$ | $31.33_{0.46}$ | $31.8_{1.11}$ |
| | | $PMI_{DC}$ Acc. | **50.4**(48.0) | $\mathbf{49.27_{2.53}}$ | $\mathbf{49.93_{3.38}}$ | $\mathbf{53.0_{3.5}}$ | $\mathbf{52.73_{2.05}}$ |
| | | Avg. PMA | 1.22 | $1.37_{0.23}$ | $1.39_{0.23}$ | $1.67_{0.19}$ | $1.9_{0.35}$ |
| | $q + \mathcal{L}_{string}$ | Accuracy | 29.2 | $27.33_{3.95}$ | $30.4_{1.11}$ | $30.6_{2.16}$ | $30.27_{0.31}$ |
| | | $PMI_{DC}$ Acc. | 43.4 | $45.13_{0.5}$ | $44.87_{1.53}$ | $44.67_{0.31}$ | $44.0_{0.8}$ |
| | | Avg. PMA | 34.49 | $59.81_{11.35}$ | $71.74_{6.01}$ | $73.27_{5.73}$ | $72.64_{3.44}$ |
| | $q + \mathcal{L}_{enum}$ | Accuracy | 24.0 | $25.27_{0.92}$ | $25.33_{1.29}$ | $26.0_{1.04}$ | $26.8_{2.09}$ |
| | | $PMI_{DC}$ Acc. | 42.6 | $42.53_{0.64}$ | $41.67_{0.31}$ | $42.2_{0.35}$ | $42.67_{0.31}$ |
| | | Avg. PMA | 67.55 | $99.04_{0.08}$ | $99.18_{0.16}$ | $99.53_{0.05}$ | $99.6_{0.04}$ |
| GPT-3 davinci (~175B) | $q$ | Accuracy | 33.6(33.2) | $33.0_{2.42}$ | $33.2_{3.03}$ | $35.73_{0.95}$ | $36.67_{0.81}$ |
| | | $PMI_{DC}$ Acc. | **57.8**(58.0) | $\mathbf{58.07_{2.58}}$ | $\mathbf{58.73_{3.58}}$ | $\mathbf{60.53_{1.45}}$ | $\mathbf{60.2_{1.04}}$ |
| | | Avg. PMA | 1.27 | $1.61_{0.43}$ | $1.61_{0.51}$ | $1.92_{0.34}$ | $2.09_{0.24}$ |
| | $q + \mathcal{L}_{string}$ | Accuracy | 34.8 | $41.67_{3.14}$ | $43.27_{2.91}$ | $46.53_{2.32}$ | $45.73_{0.7}$ |
| | | $PMI_{DC}$ Acc. | 46.6 | $49.6_{2.11}$ | $50.53_{0.95}$ | $50.53_{0.64}$ | $50.33_{1.33}$ |
| | | Avg. PMA | 43.52 | $77.98_{4.95}$ | $85.0_{3.77}$ | $83.93_{3.89}$ | $85.48_{1.97}$ |
| | $q + \mathcal{L}_{enum}$ | Accuracy | 31.2 | $39.0_{0.72}$ | $41.73_{2.55}$ | $47.47_{2.08}$ | $44.53_{1.1}$ |
| | | $PMI_{DC}$ Acc. | 42.8 | $44.07_{1.03}$ | $45.2_{0.2}$ | $47.07_{0.61}$ | $46.33_{1.36}$ |
| | | Avg. PMA | 65.75 | $98.73_{0.28}$ | $99.31_{0.24}$ | $99.57_{0.12}$ | $99.7_{0.1}$ |
| davinci-instruct-beta | $q$ | Accuracy | 36.4 | $36.4_{0.4}$ | $37.73_{2.14}$ | $38.27_{1.47}$ | $38.4_{0.72}$ |
| | | $PMI_{DC}$ Acc. | **59.6** | $\mathbf{62.27_{0.12}}$ | $\mathbf{63.07_{1.67}}$ | $\mathbf{63.6_{1.25}}$ | $\mathbf{62.67_{1.14}}$ |
| | | Avg. PMA | 1.5 | $1.79_{0.32}$ | $1.9_{0.48}$ | $2.14_{0.44}$ | $2.25_{0.28}$ |
| | $q + \mathcal{L}_{string}$ | Accuracy | 41.6 | $44.13_{5.62}$ | $46.4_{3.22}$ | $48.47_{1.03}$ | $47.73_{1.97}$ |
| | | $PMI_{DC}$ Acc. | 50.6 | $53.07_{2.01}$ | $52.8_{1.64}$ | $53.87_{1.7}$ | $52.93_{0.42}$ |
| | | Avg. PMA | 68.18 | $80.67_{5.85}$ | $84.67_{3.0}$ | $84.82_{3.03}$ | $86.37_{1.95}$ |
| | $q + \mathcal{L}_{enum}$ | Accuracy | 36.2 | $44.87_{2.19}$ | $47.87_{2.84}$ | $46.87_{2.81}$ | $48.13_{0.5}$ |
| | | $PMI_{DC}$ Acc. | 45.0 | $48.2_{1.59}$ | $48.87_{0.76}$ | $48.67_{1.03}$ | $50.4_{1.0}$ |
| | | Avg. PMA | 69.07 | $97.79_{0.45}$ | $98.92_{0.25}$ | $99.41_{0.01}$ | $99.54_{0.17}$ |
| FLAN-T5-XXL (11B) | $q$ | Accuracy | 33.2 | 33.4 | 34.6 | 35.2 | 35.8 |
| | | $PMI_{DC}$ Acc. | 58.4 | 58.4 | 60.6 | 59.8 | 61.4 |
| | | Avg. PMA | 2.28 | 2.52 | 2.49 | 2.57 | 2.53 |
| | $q + \mathcal{L}_{string}$ | Accuracy | 80.0 | 79.6 | 81.2 | 81.2 | 80.6 |
| | | $PMI_{DC}$ Acc. | 75.0 | 76.2 | 76.6 | 76.6 | 76.2 |
| | | Avg. PMA | 96.16 | 96.52 | 97.2 | 97.37 | 97.42 |
| | $q + \mathcal{L}_{enum}$ | Accuracy | **83.0** | **83.6** | **84.2** | **83.6** | **84.2** |
| | | $PMI_{DC}$ Acc. | 73.0 | 72.4 | 72.8 | 73.2 | 72.8 |
| | | Avg. PMA | 99.69 | 99.73 | 99.75 | 99.71 | 99.64 |
| text-davinci-003 | $q$ | Accuracy | 42.8 | 42.0 | 44.4 | 45.4 | 46.0 |
| | | $PMI_{DC}$ Acc. | 63.0 | 62.4 | 64.6 | 65.4 | 68.0 |
| | | Avg. PMA | 2.03 | 2.14 | 2.43 | 3.19 | 3.65 |
| | $q + \mathcal{L}_{string}$ | Accuracy | 77.0 | 77.0 | 78.0 | 80.2 | 81.0 |
| | | $PMI_{DC}$ Acc. | 77.0 | 80.2 | 80.6 | 81.6 | 81.6 |
| | | Avg. PMA | 77.21 | 93.97 | 93.28 | 93.91 | 95.49 |
| | $q + \mathcal{L}_{enum}$ | Accuracy | **80.0** | **81.6** | **81.6** | 83.0 | **83.6** |
| | | $PMI_{DC}$ Acc. | **80.4** | **81.2** | **81.8** | **84.2** | 82.6 |
| | | Avg. PMA | 99.76 | 99.96 | 99.79 | 99.83 | 99.92 |

Table 11: Full metrics for each model and prompt type on the OpenbookQA test set. See caption of Table 9 for more details. #s in parentheses are those reported in Holtzman et al. (2021), though exact model used may not be the same.

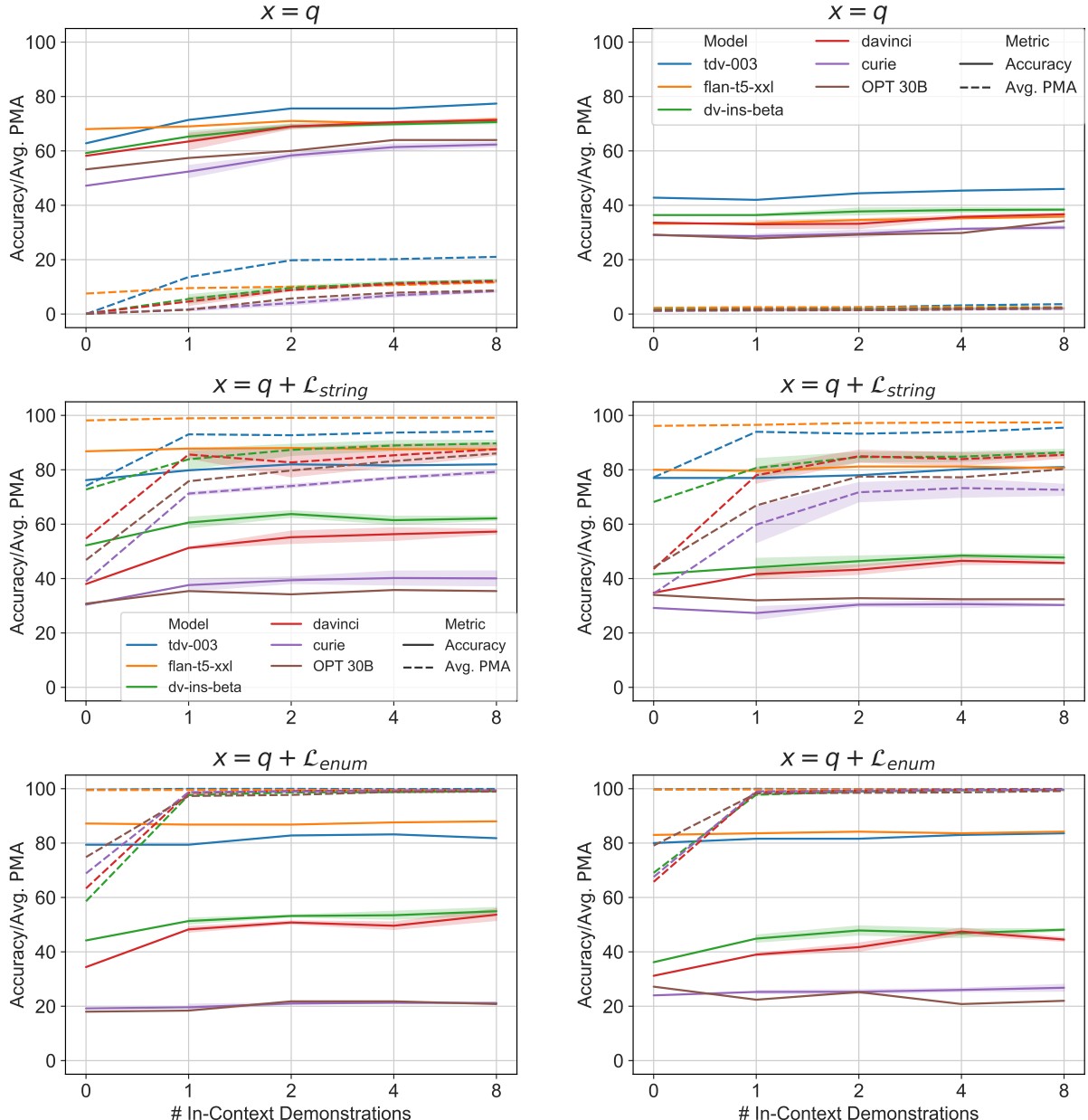

Figure 9: CommonsenseQA validation subset accuracy and average PMA as a function of number and format of in-context examples. Random accuracy is 20%. See caption of Fig. 3 for more details. Note this task is explicitly in-domain for FLAN-T5.

Figure 10: OpenbookQA test set accuracy and average PMA as a function of number and format of in-context examples. Random accuracy is 25%. See caption of Fig. 3 for more details. Note this task is explicitly in-domain for FLAN-T5.

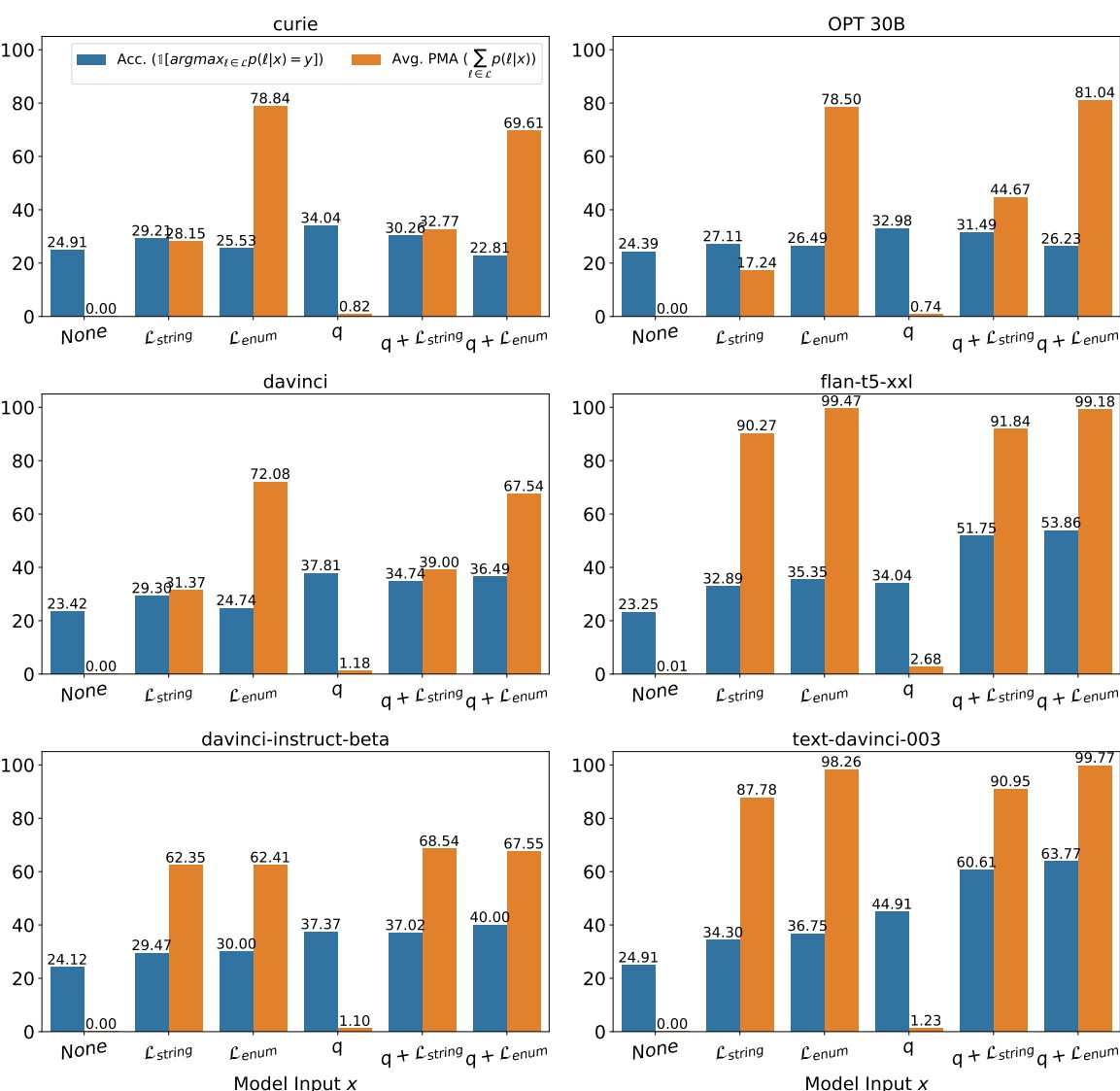

Figure 11: Zero-shot results on a subset of the MMLU test set for various LLMs. See Fig. 5 for more details.

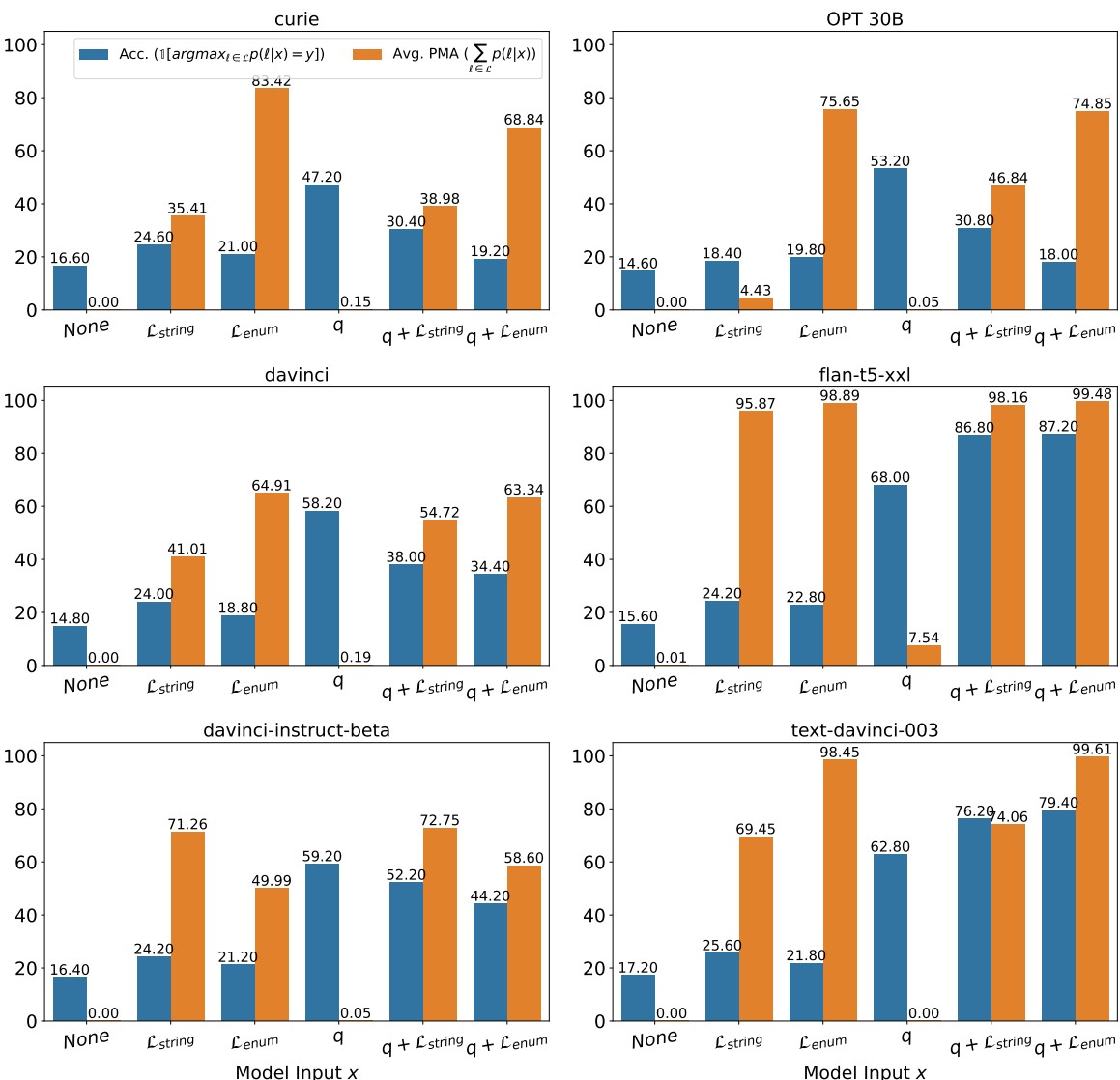

Figure 12: Zero-shot results on a subset of the CommonsenseQA validation set for various LLMs. See Fig. 5 for more details.

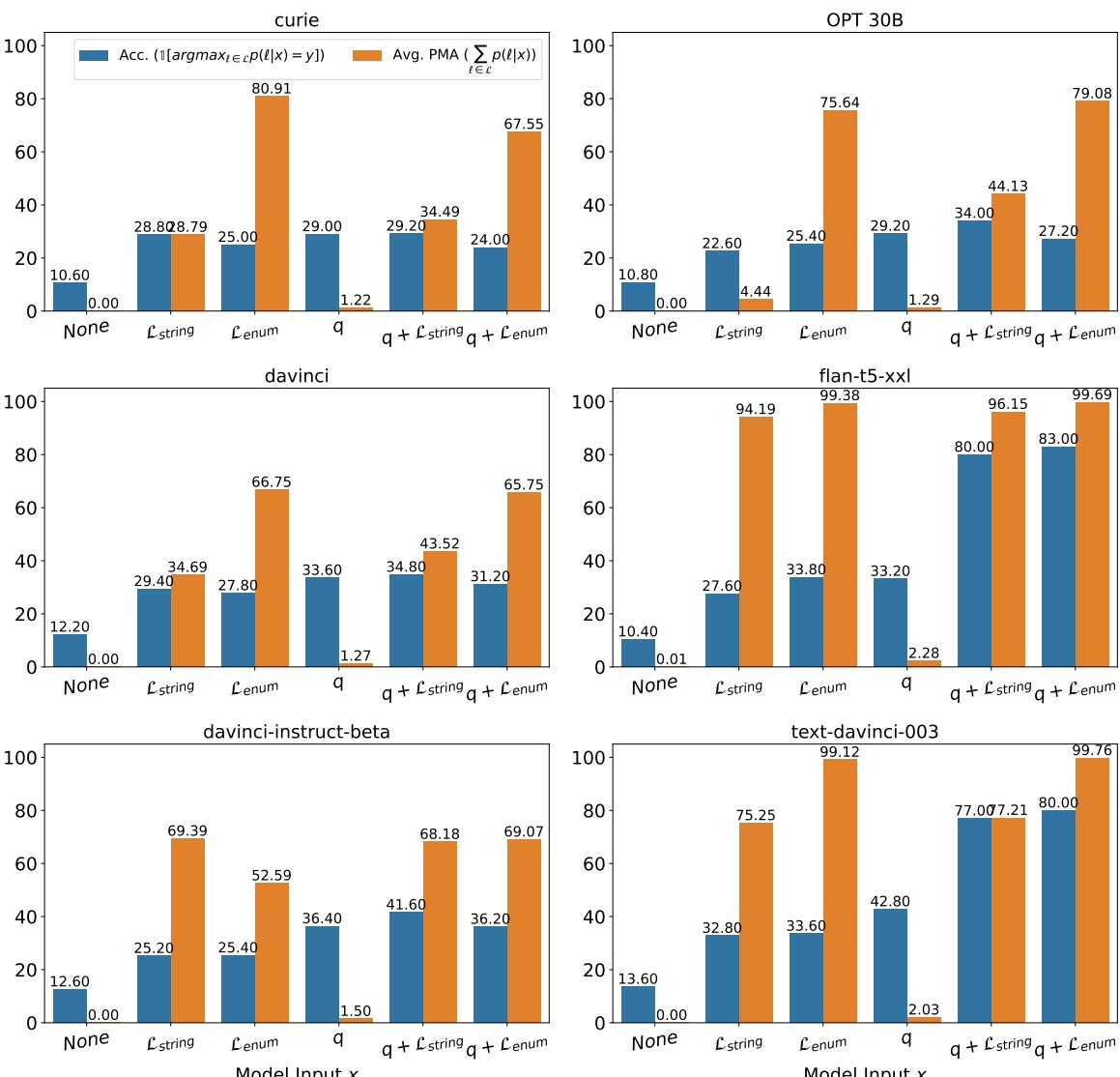

Figure 13: Zero-shot results on the OpenbookQA test set for various LLMs. See Fig. 5 for more details.