# OpenReview forum: "Increasing Probability Mass on Answer Choices Does Not Always Improve Accuracy"
_EMNLP/2023/Conference — EMNLP 2023 Main_

### Official Review · Reviewer_YTXm · 2023-08-04

**Soundness:** 4

**Excitement:**

4: Strong: This paper deepens the understanding of some phenomenon or lowers the barriers to an existing research direction.

**Paper Topic And Main Contributions:**

This paper aims to study the effects of prompt format, in-context examples on probability assigned to answer choices on different model types (e.g. vanilla language models and instruction-tuned models) and its relationship with final models' performance. Particularly, the authors introduce a new formalization of surface form competition and a quantifiable metric for understanding and improving the use of LMs for discriminative tasks.

**Questions For The Authors:**

- Question A: Figure 4 shows the lack of correlation between increases in PMA and increases in accuracy, do you consider reporting their concrete correlation, such as Spearman correlation?
- Question B: Why do you only consider simple prompt templates, do you think your findings are still applicable to complex prompts, such as chain-of-thought?


**Reasons To Accept:**

- The authors propose a mathematical formalism to quantify the "surface form competition" (SFC) hypothesis in LMs and bound its impact
- The authors introduce two simple methods to reduce the impact of SFC: (i) including choices in the prompt, and (ii) using in-context learning, thereby increasing probability mass on the given answer choices
- Finally, the authors provide some practical insights on effectively prompting different types of LMs to perform multiple-choice tasks


**Reasons To Reject:**

- This paper focuses on the task of multiple-choice question-answering, so the findings and conclusions may not be applicable to other tasks.

**Reproducibility:**

4: Could mostly reproduce the results, but there may be some variation because of sample variance or minor variations in their interpretation of the protocol or method.

**Reviewer Confidence:**

3: Pretty sure, but there's a chance I missed something. Although I have a good feel for this area in general, I did not carefully check the paper's details, e.g., the math, experimental design, or novelty.

---

> ### Author Rebuttal · Authors · 2023-08-29
>
> Thank you for appreciating our paper’s contributions!
>
> ---
>
> ***"This paper focuses on the task of multiple-choice question-answering, so the findings and conclusions may not be applicable to other tasks."***
>
> Prior work that proposed surface form competition studied only tasks with answer choices, due to the complicated nature of assessing correctness in open-ended generative settings. Multiple-choice question answering can be viewed as a general format under which to frame classification tasks, so our methods are extensible to any such task.
>
> ***"Question A: Figure 4 shows the lack of correlation between increases in PMA and increases in accuracy, do you consider reporting their concrete correlation, such as Spearman correlation?"***
>
> Good question! Here is a table with Spearman’s correlation results. Bolded are results which are statistically significant at p<0.05 for a two-sided hypothesis test:
>
> | Dataset/Model | curie | OPT 30B | davinci |davinci-instruct-beta | FLAN-T5 XXL |text-davinci-003 |
> | -------- | -------- | -------- | -------- |-------- | -------- |-------- |
> | MMLU (Fig. 4) | **-0.84** | **-0.84** | 0.45 | 0.47 | **1.00** | **0.98** |
> | CommonsenseQA (Fig. 7a) | **-0.88** | **-0.91** | -0.62 | -0.63 | **1.00** | **0.86** |
> | OpenbookQA (Fig. 7b) | -0.50 | -0.42 | **0.78** | **0.84** | **1.00** | **1.00** |
>
> We observe here further evidence for our claim that PMA and accuracy are, in fact, generally very negatively correlated in the case of curie and OPT 30B, and most positively correlated in the case of FLAN-T5 and text-davinci-003. Davinci and davinci-instruct-beta exhibit highly variable correlation. We will add these results to the paper.
>
> ***"Question B: Why do you only consider simple prompt templates, do you think your findings are still applicable to complex prompts, such as chain-of-thought?"***
>
> Good question! Holtzman et al. did not consider chain-of-thought and we followed suit.
>
>  Chain of thought works best on large and/or instruction-tuned models with few-shot examples or fine-tuning– the original paper (Wei et al. 2022) only reported accuracy improvements on models 62B parameters or larger, using 8 in-context examples. In settings where the model does not exhibit the ability to generate CoTs, and/or does not produce a clear answer choice at the end, such as "So the answer is: ____", it is complicated to measure PMA and accuracy.
>
> Our hypothesis is that forcing the generation of specific answer choices at the end of the CoT, especially when those answer choices haven’t been observed in the prompt, would likely lead to low probabilities, and therefore low PMA. We hypothesize then that the relationship between SFC and accuracy will be even more tenuous in this setting, furthering our current findings. This is an interesting question for future work.

---

### Official Review · Reviewer_owkT · 2023-08-04

**Soundness:** 4

**Excitement:**

3: Ambivalent: It has merits (e.g., it reports state-of-the-art results, the idea is nice), but there are key weaknesses (e.g., it describes incremental work), and it can significantly benefit from another round of revision. However, I won't object to accepting it if my co-reviewers champion it.

**Paper Topic And Main Contributions:**

This paper examines the "surface form competition" (SFC) hypothesis in multiple-choice questions, which refers to the distribution of probability across various surface forms with the same meanings. The authors present a mathematical formalism for a surface form that enables the quantification of SFC and the limitation of its impact. They propose simple in-context learning and a prompt-based method as strategies to reduce the effects of SFC. Furthermore, the authors analyze the correlation between task performance and the impact of SFC.

**Questions For The Authors:**

1. What does an example of input prompts without 'q' look like in Figure 5? It also raises questions about how models predict answers without the presence of a question.

**Reasons To Accept:**

1. The authors formally define the “surface form competition” (SFC) hypothesis in LM. The formulation of SFC enables bounding the SFC’s impact on an LM prediction.
2. The authors introduce two possible approaches to reduce the effect of SFC (equivalent to increasing the PMA), in-context learning and prompting with choices. Both approaches significantly increase the PMA.
3. The authors investigate the correlation between reducing SFC and multiple-choice task accuracy. Based on the observation, the authors further provide practical insights on how to use vanilla LMs and instruction-tuned models in multiple-choice settings.

**Reasons To Reject:**

Reasons To Reject
1. The actual impact of SFC in a standard multiple-choice setup may not be as significant as suggested.
- Given that multiple-choice tasks naturally include answer candidates along with the queries, incorporating these choices into a prompt appears to be a reasonable step, aligning with the "q+L_string" or "q+L_enum" configurations. However, in both setups, even zero-shot inference indicates high PMA values (close to 100 for instruction-tuned models), implying a small upper bound for SFC (smaller than 1-PMA).
- It would be beneficial to explore in more depth the real instances of "stealing" (as shown in Figure 2) that occur in multiple-choice setups, to better demonstrate the potential risks associated with SFC in multiple-choice QA.
- Performing additional experiments on NLP tasks beyond multiple-choice QA could offer a broader perspective on the potential risks of SFC and the effectiveness of reducing its impact.

2. The correlation between the impact of SFC and the performance of the multiple-choice task appears to be minimal.
- It's hard to predict that reducing the upper bound of SFC (or increasing PMA) would lead to an improvement in the performance of multiple-choice QA. The differing trends in PMA vs. Accuracy (as shown in Figure 1) could be attributed to alignment with the training input and prompt format. Vanilla LMs are trained with inputs similar to "q", while instruction-tuned models encounter prompts for classification with options (similar to "q+L") during instruction fine-tuning.
- The low PMA of "q" could be due to the vast prediction space (over |V|), as it does not restrict this space. Thus, it could be reasonably expected to observe an increase in PMA when adding L.

**Reproducibility:**

4: Could mostly reproduce the results, but there may be some variation because of sample variance or minor variations in their interpretation of the protocol or method.

**Reviewer Confidence:**

4: Quite sure. I tried to check the important points carefully. It's unlikely, though conceivable, that I missed something that should affect my ratings.

---

> ### Author Rebuttal · Authors · 2023-08-29
>
> Thank you for your detailed response. There seems to be some substantial misunderstandings about our paper that we hope to clarify. Our stance is not to agree with Holtzman et al. (2021) that SFC has a large effect, but rather that SFC is less important/less predictive of task performance than previously assumed. We believe you may feel differently about the paper in this light. If anything is still unclear, we are happy to discuss further.
>
> ---
>
> ***The actual impact of SFC in a standard multiple-choice setup may not be as significant as suggested.***
>
> Indeed, this is one of our main findings! Prior work (Holtzman et al., 2021) suggested that SFC has a lot of impact, but we find and report that this actually isn't the case, as you also observe.
>
> The original SFC paper (Holtzman et al) drew conclusions from their experiments based on **accuracy alone**. When they observed increased accuracy in a standard multiple-choice setup using probability normalization (their proposed method to combat SFC), they concluded that 1) SFC existed, 2) it had a substantial negative impact on accuracy, and 3) that negative impact was reduced by using their method. In lines 73-77 of our paper, we explain why we think this isn’t a valid approach. As an alternative, we propose a metric (PMA) to bound SFC’s possible impact, so we can test the impact of SFC on accuracy more directly.
>
> As we note in lines 93-103, one of our main findings is very much aligned with your view — namely, that the actual impact of SFC in a standard multiple-choice setup is often not as significant as suggested by prior work. For example in the $q + \mathcal{L}_{enum}$ case in Figure 1, we observe that for some models, PMA is high but accuracy is worse than when PMA is low.
>
> ***Given that multiple-choice tasks naturally include answer candidates along with the queries, incorporating these choices into a prompt appears to be a reasonable step, aligning with the "q+L_string" or "q+L_enum" configurations. However, in both setups, even zero-shot inference indicates high PMA values (close to 100 for instruction-tuned models), implying a small upper bound for SFC (smaller than 1-PMA).***
>
> Exactly, the addition of answer choices in the prompt ("q+L_string" or "q+L_enum") is very natural and we find that this leads to high PMA values (and thus low SFC), as you note. This again bolsters our main claim (aligned with your intuition) that reduction in SFC isn’t well correlated with an increase in accuracy.
>
> ***It would be beneficial to explore in more depth the real instances of "stealing" (as shown in Figure 2) that occur in multiple-choice setups, to better demonstrate the potential risks associated with SFC in multiple-choice QA.***
>
> As mentioned above, previous work has already explained and claimed risks from SFC; our goal here is to explain that SFC is not really a big issue!
>
> ***The correlation between the impact of SFC and the performance of the multiple-choice task appears to be minimal.***
>
> Indeed, this is our main finding, as clarified above.
>
> ***It's hard to predict that reducing the upper bound of SFC (or increasing PMA) would lead to an improvement in the performance of multiple-choice QA.***
>
> This is exactly why we measure both quantities (PMA and accuracy) and correlation between them (for example, in Figure 4).
>
> ***The differing trends in PMA vs. Accuracy (as shown in Figure 1) could be attributed to alignment with the training input and prompt format. Vanilla LMs are trained with inputs similar to "q", while instruction-tuned models encounter prompts for classification with options (similar to "q+L") during instruction fine-tuning.***
>
> We very much agree with you! Alignment with the training input and prompt format is one of our main hypotheses for our finding that prompt format directly impacts both PMA and accuracy– we discuss this in lines 432-433, 437-440, 484-485, & 488-491.
>
> ***The low PMA of "q" could be due to the vast prediction space (over |V|), as it does not restrict this space. Thus, it could be reasonably expected to observe an increase in PMA when adding L.***
>
> Exactly. We call this the "under-constrained output space" issue in lines 85-88, and that’s why we felt that adding $\mathcal{L}$ was a reasonable solution that was far more straightforward than the complicated and indirect solution proposed in prior work (probability normalization).
>
> ***Question 1: What does an example of input prompts without 'q' look like in Figure 5?***
>
> Good question- we explain this in Appendix A.4 for the "None" case and will add a reference to this section in Figure 5’s caption. It is the same as the prompts in Sec. 6.3, minus the question:
>
> $x = None$:
>
> "? "
>
> $x=\mathcal{L}_{string}$:
>
> "answer choices: snacks, naps, kites, or warmth
>
> The correct answer is:"
>
> $x=\mathcal{L}_{enum}$:
>
> "Choices:
>
> A: snacks
>
>  B: naps
>
>  C: kites
>
>  D: warmth
>
> Answer:"
>
> ***It also raises questions about how models predict answers without the presence of a question.***
>
> Even though the question is not specified, we can still check the probability mass of the answer choices, which will likely be near 0, hence the 0.00 PMA value in the figure. Holtzman et al. use the same methodology to compute the denominator in $\mathrm{PMI}_{\mathrm{DC}}$. When we add answer choices to the context, even without the question, the probability mass on the choices will go up, though accuracy should stay about the same (near random). This experiment further supports the intuition that simply increasing PMA (and thus reducing SFC) is not an effective way to increase model accuracy.

---

### Official Review · Reviewer_3J8R · 2023-08-09

**Soundness:** 4

**Excitement:**

4: Strong: This paper deepens the understanding of some phenomenon or lowers the barriers to an existing research direction.

**Paper Topic And Main Contributions:**

The paper studies the problem of how to measure and reduce surface form competition for multiple choice QA tasks, and investigates the relationship between surface form competition and task accuracy.

They propose a novel metric PMA to measure the upper bound of surface form competition (SFC), and illustrate that prompting the model to output option letters instead of the answer string with in-context examples help reduce SFC. They also conduct experiments on multiple QA datasets to illustrate this and brings up the conclusion that reducing SFC doesn't always improve model performance on multiple choice questions.

**Reasons To Accept:**

1. The paper is very well-written and easy to follow, with all the details explained clearly.
2. The authors proposes an interesting approach to think of and measure surface form competition (SFC) and is well justified.
3. They conduct well-designed experiments to show the relationship between SFC and accuracy, and demonstrates that lower SFC doesn't necessarily lead to better accuracy, which could be insightful for future work.

**Reasons To Reject:**

It's well known that that can be many factors that influence the final task performance, e.g., relevance and/or order of in-context examples, etc. The conclusion that lower SFC doesn't always lead to better accuracy sounds a bit plain, because there are many angles to look at for understanding the task performance. For example, prompting the model to output option letters doesn't only change SFC, but many other things (additional reasoning required by mapping answer string to letters, prompt style, etc). I'm not sure how much insights this analysis brings to the community.

**Reproducibility:**

4: Could mostly reproduce the results, but there may be some variation because of sample variance or minor variations in their interpretation of the protocol or method.

**Reviewer Confidence:**

3: Pretty sure, but there's a chance I missed something. Although I have a good feel for this area in general, I did not carefully check the paper's details, e.g., the math, experimental design, or novelty.

---

> ### Author Rebuttal · Authors · 2023-08-29
>
> Thank you for finding our methodology interesting and well-justified, our experiments well-designed, our writing clear and easy to follow, and our findings insightful. We answer your critiques below.
>
> ***"It's well known that that can be many factors that influence the final task performance, e.g., relevance and/or order of in-context examples, etc."***
>
> These are important points that we address in our paper. We are careful to rule out major confounds in our experiments, specifically, variance from in-context example ordering. In Section 6.3 L423 and Section 7.3 L554-557, we mention that we run the experiments over multiple random seeds used to draw in-context examples from the training set, and that the resulting variance caused by this is negligible (please see Tables 8-10 for the standard error statistics). We study task format as an independent variable in our experiments, and are the first to study its effect, as well as the effect of in-context examples, on surface form competition, measured by our proposed metric (PMA).
>
> ***"The conclusion that lower SFC doesn't always lead to better accuracy sounds a bit plain, because there are many angles to look at for understanding the task performance. For example, prompting the model to output option letters doesn't only change SFC, but many other things (additional reasoning required by mapping answer string to letters, prompt style, etc). I'm not sure how much insights this analysis brings to the community."***
>
> Our conclusions are more nuanced than a simple "SFC does/doesn’t impact performance", and we believe that this reflects the complex reality of the factors at play here. Our conclusions and hypotheses about when (not) to apply probability normalization and multiple choice formats based on model type are both practically useful and provide further understanding of the issue (as pointed out by other reviewers, e.g., oE9K). Though there are many angles for looking at task performance, we assert that it is worthwhile to isolate specific angles for close study such as SFC, especially considering the attention that was brought to it by prior work. We **invalidate** the hypothesis from prior work that reducing SFC improves the performance of vanilla LMs on multiple-choice questions.
>
> If we understand your concern properly, it may be a misunderstanding of the scope of our claims. In the example you give, we are **not** saying that SFC is the **only** thing that changes when one prompts a model to output option letters, but we are saying that the choice of prompt format impacts **both** accuracy and SFC, and (unintuitively) not necessarily in the same direction.

---

### Official Review · Reviewer_oE9K · 2023-08-11

**Soundness:** 4

**Excitement:**

3: Ambivalent: It has merits (e.g., it reports state-of-the-art results, the idea is nice), but there are key weaknesses (e.g., it describes incremental work), and it can significantly benefit from another round of revision. However, I won't object to accepting it if my co-reviewers champion it.

**Paper Topic And Main Contributions:**

The paper studies the relation between surface form competition (SFC) and classfication accuracy of language models. To quantify SFC, the author uses total probability mass on answer choices. This quantity is then compared with model accuracy under different prompt designs, such as whether to include the choices as a string.

The main observation of the paper is that increasing total probability mass on answer choices does not necessarily increase accuracy. The main takeaway is (quote)

The best way to use vanilla LMs in multiple-choice settings is to provide a string prompt without answer choices and apply probability normalization. For instruction-tuned models, on the other hand, answer choices should be shown and in an enumerated prompt format, and probability normalization should not be used

**Reasons To Accept:**

- The result is interesting: including multiple choices in the prompt can hurt performance of vanilla LMs while improving instruction-tuned LMs.
- Experimented with 6 LMs, 3 prompt settings and 3 tasks.
- The practical insights are concrete and executable.

**Reasons To Reject:**

In Figure 6(b), adding answer choices to the prompt and apply probability normalization hurts Davinci performance significantly. This phenomenon seems to contradict the insights proposed in the paper, that vanilla LMs benefits from probability normalization. With the Curie model (a smaller version of Davinci) agreeing with the insight, this may suggest that larger model size can invalidate part of this paper's hypothesis.

**Reproducibility:**

4: Could mostly reproduce the results, but there may be some variation because of sample variance or minor variations in their interpretation of the protocol or method.

**Reviewer Confidence:**

2: Willing to defend my evaluation, but it is fairly likely that I missed some details, didn't understand some central points, or can't be sure about the novelty of the work.

---

> ### Author Rebuttal · Authors · 2023-08-28
>
> Thank you for appreciating the interestingness of our findings, the extensiveness of our experiments, and the concreteness and executability of our practical takeaways! We address your concern below. Please let us know if you have any other questions/comments.
>
> ***"In Figure 6(b), adding answer choices to the prompt and apply probability normalization hurts Davinci performance significantly. This phenomenon seems to contradict the insights proposed in the paper, that vanilla LMs benefits from probability normalization. With the Curie model (a smaller version of Davinci) agreeing with the insight, this may suggest that larger model size can invalidate part of this paper's hypothesis."***
>
> We’d like to clarify the main insight proposed in the paper: vanilla LMs benefit from applying probability normalization ***and*** not showing the answer choices (lines 573-576). This is supported by the davinci results:
> - The "davinci-q" line in Figure 6(b): when answer choices are not shown, probability normalization increases accuracy.
> - Figure 6(a): here, we compute the difference between the best performing prompt with probability normalization vs. without. The "davinci" line shows that on all 3 datasets, probability normalization leads to a 1.5-15 point increase in accuracy.
> - Tables 8-10 (Appendix) show the complete accuracies used to produce Figure 6. The prompt that leads to these 1.5-15 point gains is not showing the answer choices (prompt $q$) and using probability normalization.

---

### Meta-Review · Area_Chair_1YUM · 2023-09-15

**Recommendation:** 4

**Metareview:**

This paper delves into the concept of "surface form competition" (SFC) and its correlation with the classification accuracy of language models, especially in multiple-choice QA tasks. The research introduces a metric, PMA, to determine the upper boundary of SFC, thereby providing insights into its potential influence. Through a series of experiments on varied QA datasets, the study indicates that increased probability mass on answer choices, which quantifies SFC, doesn't always elevate model accuracy.

The main strengths of the paper are:
1) The paper is well-written and easy to follow, formal definition of the task is given
2) The proposed approach is interesting and provides practical insights on how to use vanilla LMs and instruction-tuned models in multiple-choice settings.
3) The experiment is well design and comprehensive

The main concern from reviewers is  The results are less conclusive, some observed phenomena contradict the insights proposed in the paper and may not be applicable to other tasks.

---

### Decision · Program_Chairs · 2023-10-07

**Decision:**

Accept-Main

**Comment:**

This paper delves into the concept of "surface form competition" (SFC) and its correlation with the classification accuracy of language models, especially in multiple-choice QA tasks. The research introduces a metric, PMA, to determine the upper boundary of SFC, thereby providing insights into its potential influence. Through a series of experiments on varied QA datasets, the study indicates that increased probability mass on answer choices, which quantifies SFC, doesn't always elevate model accuracy.

The main strengths of the paper are:
1) The paper is well-written and easy to follow, formal definition of the task is given
2) The proposed approach is interesting and provides practical insights on how to use vanilla LMs and instruction-tuned models in multiple-choice settings.
3) The experiment is well design and comprehensive

The main concern from reviewers is  The results are less conclusive, some observed phenomena contradict the insights proposed in the paper and may not be applicable to other tasks.